# Variation of size-segregated particle number concentrations in winter Beijing

Ying Zhou[1], Lubna Dada[1,2]*, Yiliang Liu[3], Yueyun Fu[4], Juha Kangasluoma[1,2], Tommy Chan[1], Chao Yan[2], Biwu Chu[2], Kaspar R Daellenbach[2], Federico Bianchi[2], Tom Kokkonen[2], Yongchun Liu[1], Joni Kujansuu[1,2], Veli-Matti Kerminen[2], Tuukka Petäjä[2], Lin Wang[3], Jingkun Jiang[4], Markku Kulmala[1,2]*

[1]Aerosol and Haze Laboratory, Beijing Advanced Innovation Center for Soft Matter Science and Engineering, Beijing University of Chemical Technology, Beijing, China

[2]Institute for Atmospheric and Earth System Research / Physics, Faculty of Science, University of Helsinki, Finland

[3]Shanghai Key Laboratory of Atmospheric Particle Pollution and Prevention (LAP[3]), Department of Environmental Science & Engineering, Jingwan Campus, Fudan University, Shanghai 200438, China

[4]School of Environment, Tsinghua University, Beijing, China

*Correspondences are to Lubna Dada: lubna.dada@helsinki.fi and Markku Kulmala: markku.kulmala@helsinki.fi

## Abstract

The spatial and temporal variability of the number size distribution of aerosol particles is an indicator of the dynamic behavior of Beijing's atmospheric cocktail. This variation reflects the strength of different primary and secondary sources, such as traffic and new particle formation, as well as the main processes affecting the particle population. In this paper, we report size-segregated particle number concentrations observed at a newly-developed Beijing station during the winter of 2018. Our measurements covered particle number size distributions over the diameter range of 1.5 nm-1 µm (cluster mode, nucleation mode, Aitken mode and accumulation mode), thus being descriptive of a major fraction of the processes taking place in the atmosphere of Beijing. Here we focus on explaining the concentration variations in the observed particle modes by relating them to the potential aerosol sources and sinks, and on understanding the connections between these modes. We considered haze days and new particle formation event days separately. Our results show that during the new particle formation (NPF) event days increases in cluster mode particle number concentration were observed, whereas during the haze days high concentrations of accumulation mode particles were present. There was a tight connection between the cluster mode and nucleation mode on both NPF event and haze days. In addition, we correlated the particle number concentrations in different modes with concentrations of trace gases and other parameters measured at

our station. Our results show that the particle number concentration in all the modes
correlated with $NO_x$, which reflects the contribution of traffic to the whole sub-micron
size range. We also estimated the contribution of ion-induced nucleation in Beijing, and
found this contribution to be negligible.
**1    Introduction**
Atmospheric aerosols are the main ingredient of China's pollution cocktail (Kulmala
2015). Aerosols have gained increasing attention due to their effects on human heath,
climate and visibility (Lelieveld et al., 2015, IPCC 2007). Currently, air quality
standards for cities in China consider particle mass instead of number concentration
(WHO, 2000), which may ignore the potential adverse effect of ultra-fine particles on
health (diameter less than 100 nm). It has been shown that ultra-fine particles can
penetrate deep into the respiratory tract, ending up to the blood circulation, which
allows them to deposit into the brain (Oberdörster et al., 2004). Indeed, studies have
pointed out that ultra-fine particles, which contribute to a negligible fraction of the mass
concentration, dominate the total number concentration in urban areas (von Bismarck-
Osten et al., 2013; Wehner et al., 2004; Wu et al., 2008). Due to their high concentrations,
ultrafine particles' toxicological effects are enhanced by their large total surface area
(Kreyling et al., 2004).
Apart from their health effects, the temporal and spatial variation of particle number
concentrations of different sizes is a good indicator of the strength of their emission
sources. Aerosols are emitted either directly as primary particles, such as sea salt or dust
particles as a result of natural phenomena (Solomos et al., 2011), or they can be formed
through new particle formation (Kulmala, 2003; Kulmala et al., 2004; Kulmala et al.,
2013; Kerminen et al., 2018; Chu et al., 2019). Newly formed particles can grow up
diameters of 20-100 nm within a day (Kulmala et al., 2004), and they have been found
to contribute to a major fraction of the global cloud condensation nuclei population
(CCN), thus indirectly affecting the climate (Kerminen et al., 2012). For all
aforementioned reasons, and in order to form a collective and complete picture about
atmospheric aerosol particles to understand their origin and potential impacts at a
specific location, the whole size distribution of these particles needs to be studied.
Recently, due to urbanization and increased population, megacities have increased their
contribution to atmospheric aerosol pollution massively (Baklanov et al., 2016).
Interestingly, more people live inside eastern Asia (specifically, China and India) than
outside this region (https://www.unfpa.org/swop). Therefore, it is important to study the
contributions of different sources to size-segregated number concentrations in order to
inspire policy makers and the public on measures that need to be taken in order to reduce
particulate pollution. Many studies in various cities in China have tackled this topic.

For instance, two-years of observations of particle number size distributions at a site in northern Beijing reported that traffic emissions were the major source of nucleation (3-20 nm) and Aitken (20-100 nm) mode particles in urban Beijing (Wang et al., 2013). On the other hand, research conducted in western downtown of Nanjing reported that local new particle formation events were the main contributors of both nucleation (5-20 nm) mode and CCN particle populations (Dai et al., 2017). Measurements of nucleation mode particle concentrations in urban Hong Kong reported the dominant contribution of combustion sources to the nucleation mode (5.5-10 nm) (Wang et al., 2014a), whereas observations in urban Guangzhou found that accumulation and secondary transformation of particles were the main reasons for high concentrations of accumulation mode particles (100-660 nm) (Yue et al., 2010). However, only a few studies in China have reported measurements of cluster mode (sub-3 nm) particles and related them to new particle formation events (Cai et al., 2017;Xiao et al., 2015;Yao et al., 2018;Yu et al., 2016).

The observation of sub-3 nm particles and ions has been made possible by recent major developments in instrumentations, such as the particle size magnifier (PSM) (Vanhanen et al., 2011), diethylene glycol-based scanning mobility particle sizer (DEG-SMPS) (Jiang et al., 2011) and Neutral Cluster and Air Ion Spectrometers (NAIS) (Manninen et al., 2016; Mirme et al., 2007).

In complicated environments like Beijing, it is very hard to relate each particle mode to a specific source. Indeed, several sources could contribute to aerosol particles in the same size range. For instance, cluster mode particles mainly originate from secondary gas-to-particle transformation processes (Kulmala et al. 2013), although recently also traffic has been identified as a source for these particles (Rönkkö et al., 2017). While cluster mode particles can grow into the Aitken mode , also other sources like traffic contribute to this mode, making the source identification of the Aitken mode complicated (Pirjola et al., 2012). Various anthropogenic activities and biogenic processes contribute to accumulation mode particle sizes. Thus, correlating trace gases and aerosol concentrations of different sizes during different time periods help narrowing down these aerosol sources.

In this study, we analyzed the number concentration of four sub-micron aerosol modes: cluster mode (sub-3 nm), nucleation mode (3-25 nm), Aitken mode (25-100 nm), and accumulation mode (100-1000 nm). Our aims were i) to investigate the number concentration variations of size-segregated aerosol number concentrations for each mode, ii) to explore the relationships between the different modes under different atmospheric conditions, iii) to connect the number size distribution modes with multiple trace gases ($NO_x$, $SO_2$, CO and $O_3$) and $PM_{2.5}$ (particulate matter with aerodynamic diameter less than 2.5 μm), and iv) to quantify the contribution of NPF and haze formation to different particle modes in winter time in Beijing. Our work increases understanding on the sources of the different sized particles in Beijing, China, and the

113   work complements studies in other megacities.

## 2   Materials and Methods

### 2.1   Description of SMEAR Beijing station

Beijing, as the capital of China, accommodates more than 20 million people within 16.8 thousand square kilometers and only 1.4 thousand square kilometers for urban areas, with an expanding economic activity, construction and industry. Beijing, as one of the largest megacities in the world, is located in the Northern Chinese Plain, and is one of the most industrialized regions in China. Mountains surround Beijing from the west, north and north-west.

For our study, we analyzed data collected at the newly-developed station which is part of the Aerosol and Haze Laboratory in Beijing. The urban station follows the concept of Station for Measuring Ecosystem and Atmospheric Relations (SMEAR) (Hari and Kulmala, 2005). Our station is located on the western campus of Beijing University of Chemical Technology (BUCT). It is constructed on the fifth floor of the teaching building on the campus. The sampling lines extend to the rooftop of the building around 20 m above the ground level, going directly through windows for selected instruments. The station represents a typical area in urban Beijing subject to pollution sources, such as traffic, cooking and long-range transport of pollution. The campus is surrounded by highways and main roads from the east (3rd ring main road), north (Zizhu road) and south-east (Zizhu Bridge). From the east, west and south, the campus is surrounded by residential and commercial areas.

Measurements at SMEAR Beijing started on 16 January, 2018 (Lu et al., 2018). Our measurements continued until present, except during the necessary instruments maintenance and unavoidable factors such as power cuts. The data included in this study were collected between 16 January and 15 March 2018, being representative of Beijing winter conditions.

### 2.2   Instrumentation

For a comprehensive measurement of particles, a full set of particle measuring instrumentation was operated. First, a nano-condensation nucleus counter system (nCNC) consisting of a Particle Sizer Magnifier (PSM, model A10, Airmodus Oy, Finland) and butanol condensation particle counter (CPC) (model A20, Airmodus Oy, Finland) measured the number concentration of small clusters or particles of 1.2-2.5 nm in mobility diameter (Vanhanen et al., 2011). To minimize the sampling losses, the

PSM was sampling horizontally through a window to the north through a short stainless steel sampling inlet extending ~1.2 m outward from the building. The length of the sampling tube was 1.33 m and its inner diameter was 0.8 cm. To further improve the sampling efficiency, a core sampling tube (Kangasluoma et al., 2016) was utilized. The total flow rate was 7.5 liters per minute (lpm), from which 5 lpm was used as a transport flow while the nCNC sample flow rate was 2.5 lpm. In the operation of the PSM, the saturator flow rate scanned from 0.1 to 1.3 lpm and scanned back from 1.3 to 0.1 lpm within 240 s. We averaged the data over 3 scans to make it smoother, and therefore the time resolution of PSM data was 12 minutes. The data were inverted with a kernel function method. When comparing the particle number concentrations obtained with the expectation-maximization method, the cluster mode particle number concentration was, on average, twice higher on the NPF event days and eleven times higher on the haze days (Cai et al., 2018). Therefore, there is some uncertainty in the reported cluster mode particle concentrations.

A particle size distribution (PSD) system measured the particle number size distribution in the size range of 3 nm-10000 nm (Liu et al., 2016). It included a nano-scanning mobility particle sizer (nano SMPS, 3-55 nm, mobility diameter), a long SMPS (25-650 nm, mobility diameter) and an aerodynamic particle sizer (APS, 0.55 μm-10 μm, aerodynamic diameter). The PSD system sampled from the rooftop using a 3-m-long sampling tube. A cyclone that removed particles larger than 10 μm was added in front of the sample line. The time resolution of PSD system data was 5 minutes.

A Neutral Cluster and Air Ion Spectrometer (NAIS, model 4-11, Airel, Estonia) measured number size distributions of particles (2.5-42 nm, mobility diameter) and ions (0.7-42 nm, mobility diameter) (Manninen et al., 2016; Mirme and Mirme, 2013). It switched between detecting either naturally charged ions or total particles (including the uncharged fraction) with unipolar charging. It measured 2 min in the neutral mode, 2 min in the ion mode and then offset for 30 seconds for every measurement cycle. The NAIS was sampling horizontally from the north window. The copper sampling tube with an outer diameter of 4 cm extended 1.6 m outside the window. To increase the sampling efficiency, the sampling flow rate was 54 lpm.

The trace gas monitors measured carbon monoxide (CO), sulfur dioxide ($SO_2$), nitrogen oxides ($NO_x$) and ozone ($O_3$) concentrations with Thermo Environmental Instruments models 48i, 43i-TLE, 42i, 49i, respectively. They all sampled through a common inlet through the roof of the building. The length of the sampling tube was approximately 3 m. The time resolution of CO, $NO_x$, and $O_3$ data were 5 minutes, whereas the time resolution of $SO_2$ data was 1 hour before 22 January, 2018, and 5 minutes after that.

The $PM_{2.5}$ data were obtained from the nearest national monitor station, Wanliu station, around 3 km north of our station. The $PM_{2.5}$ data from Wanliu station compared nicely with the $PM_{2.5}$ data from three other adjacent national stations. The time resolution of

the PM$_{2.5}$ data was 1 hour, and these data were recorded every hour. Detailed
information is reported in Cao et al. (2014).
We measured the relative humidity (RH, %), visibility (km), wind speed (m/s) and wind
direction (˚) from a weather station on the roof of our station.
When data sets having different time resolutions were used, we chose the smallest time
resolution as the common time resolution. Data with higher time resolutions were
merged to the common time resolution by taking median numbers between two time
points of the new time series.
**2.3 NPF events and haze days classification**
We classified days into "NPF event days" and "haze days". The days that did not fit
either of these two categories were marked as "Other days", and they were excluded
from our future analysis unless otherwise specified. We observed 28 NPF event days
and 24 haze days in total. Table 1 describes the specific calendar of events with the
aforementioned categories of days.
We identified the NPF event days following the method introduced in (Dal Maso et al.,
2005), which requires an appearance of a new mode below 25 nm and that the new
mode shows signs of growth for several hours (Dal Maso et al., 2005; Kulmala et al.,
2012). Haze events were identified as having a visibility less than 10 km and ambient
relative humidity below 80% (China Meteorological Administration). Individual days
were classified as haze days when the haze event lasted for at least 12 consecutive hours.
During our study periods, there was no overlap between the NPF events and haze days,
as these two phenomena never occurred simultaneously. While the NPF events
appeared right after sunrise and lasted for several hours, the haze events did not have
any specific time of appearance but lasted from a few hours up to several days.
The particle number size distribution was divided into 4 modes according to their
diameter: cluster mode (sub-3 nm), nucleation mode (3-25 nm), Aitken mode (25-100
nm), and accumulation mode (100-1000 nm). We calculated cluster mode particle
number concentrations using Particle Size Magnifier (PSM) data, nucleation mode
particle number concentration using Neutral Cluster and Air Ion Spectrometer (NAIS)
particle mode data, and Aitken and accumulation mode particle number concentrations
using Particle Size Distribution (PSD) system data. The Particle Size Distribution
system (PSD) and Neutral Cluster and Air Ion Spectrometer (NAIS) had an overlapping
particle size distribution over the mobility diameter range of 3-42 nm. As shown in
Figure S1, total particle number concentrations from the NAIS and PSD system
correlated well with each other on both NPF event days ($R^2$ was 0.92) and haze days
($R^2$ was 0.90) in the overlapping size range. The slopes between the total particle
number concentration from the PSD system and that from the NAIS were 0.90 and 0.85
on the NPF event days and haze days, respectively. The particle number size
distribution in the overlapping size range of the NAIS and PSD system matched well
on both NPF event days and haze days as shown in Figure S2.
Moreover, since new particle formation events were only observed during daytime in
Beijing, our analysis concentrated mostly on the time period 8:00 to 14:00, unless
specified otherwise.
## 2.4    Parameter calculation
### 2.4.1 Calculation of the growth rate
The growth rates of cluster and nucleation mode particles were calculated from positive
ion data and particle data from Neutral Cluster and Air Ion Spectrometer (NAIS),
respectively, by using the appearance time method introduced by Lehtipalo et al. (2014).
In this method, the particle number concentration of particles of size $dp$ is recorded as
a function of time, and the appearance time of particles of size $dp$ is determined as the
time when their number concentration reaches 50% of its maximum value during new
particle formation (NPF) events.
The growth rates (GR) were calculated according to:
$$GR = \frac{dp_2 - dp_1}{t_2 - t_1}$$
(1)

where $t_2$ and $t_1$ are the appearance times of particles with sizes of $dp_2$ and $dp_1$
respectively. Figure S3 shows an example of how this method was used.
### 2.4.2 Calculation of the coagulation sink
The coagulation sink (CoagS) was calculated according to the equation (2) introduced
by Kulmala et al. (2012):
$$CoagS_{dp} = \int K(dp, d'p)n(d'p)dd'p \cong \sum_{d'p=dp}^{d'p=max} K(dp, d'p)N_{d'p}$$

(2)

where $K(dp, d'p)$ is the coagulation coefficient of particles with sizes of $dp$ and
$d'p$, $N_{d'p}$ is the particle number concentration with size of $d'p$.

### 2.4.3 Calculation of the formation rate

The formation rate of 1.5-nm particles ($J_{1.5}$) was calculated using particle number concentrations measured with a Particle Sizer Magnifier (PSM). The formation rate of 1.5-nm ions ($J_{1.5}^{\pm}$) was calculated using positive and negative ions data from the Neutral Cluster and Air Ion Spectrometer (NAIS) as well as PSM data. The upper limit used was 3 nm. The values of $J_{1.5}$ and $J_{1.5}^{\pm}$ were calculated following the methods introduced by Kulmala et al. (2012) with equation (3) and equation (4), respectively:

$$J_{dp} = \frac{dN_{dp}}{dt} + CoagS_{dp} \cdot N_{dp} + \frac{GR}{\Delta dp} \cdot N_{dp} \tag{3}$$

where $CoagS_{dp}$ is the coagulation sink in the size range of $[dp, dp + \Delta dp]$ and GR is the growth rate.

$$J_{dp}^{\pm} = \frac{dN_{dp}^{\pm}}{dt} + CoagS_{dp} \cdot N_{dp}^{\pm} + \frac{GR}{\Delta dp} \cdot N_{dp}^{\pm} + \alpha \cdot N_{dp}^{\pm} \cdot N_{<dp}^{\mp} - \chi N_{dp} \cdot N_{<dp}^{\pm} \tag{4}$$

The fourth and fifth terms on the right hand side of equation (4) represent ion-ion recombination and charging of neutral particles by smaller ions, respectively, $\alpha$ is the ion-ion recombination coefficient and $\chi$ is the ion-aerosol attachment coefficient.

## 3    Results and discussion

### 3.1 General character of particle modes and trace gases

### 3.1.1 Sub-micron particles and PM2.5

Particle number concentrations of different modes varied depending on the period, as shown in Figure 1. We observed that the cluster and nucleation mode particle concentrations were the highest on the NPF event days. In fact, the cluster and nucleation mode particles dominated the total particle number concentration with an average contribution of 96% (Figure 2). On the haze days, the average contribution levels of the four modes were about equal. Aitken and accumulation mode particles contributed to 52% of the total particle number concentration on the haze days, as compared to 4% on the NPF event days.

On the haze days, we observed a surprising concentration of cluster mode particles in spite of the high concentrations of Aitken and accumulation particles. Since large particles are expected to efficiently scavenge clusters and smallest growing particles by coagulation (Kerminen et al., 2001; Kulmala et al., 2017), this is indicative of either airborne cluster formation (Kulmala et al., 2007) or vehicular emissions of clusters and nucleation mode particles (e.g. Rönkkö et al., 2017) during haze. The ratio between

nucleation mode and cluster mode particle median number concentration was close to unity (0.84), which might indicate their common source on haze days, in comparison to the smaller ratio of 0.3 during the NPF days. It is therefore likely that the primary particles dominated the nucleation mode on the haze days, while the growth of cluster mode particles into nucleation mode explains the nucleation mode particles on NPF days.

The median concentrations of Aitken and accumulation mode particles were 16000 cm$^{-3}$ and 17500 cm$^{-3}$, respectively, on the haze days and 8240 cm$^{-3}$ and 1670 cm$^{-3}$, respectively, on the NPF event days. Overall, these concentrations were a factor of 2.1 and 10.5 times higher on the haze days than on the NPF event days. The PM$_{2.5}$ mass concentration was clearly higher on the haze days compared with the NPF event days (Figure 3). The PM$_{2.5}$ mass concentration in urban areas is dominated by accumulation mode particles, with a clearly smaller a contribution by ultrafine (cluster, nucleation and Aitken mode) particles (Feng et al., 2010).

### *3.1.2 Trace gases*

In this work, we considered four trace gases (SO$_2$, CO, NO$_x$ and O$_3$) in our analysis (Figure 4), as these compounds are most commonly used to evaluate air quality and pollution sources in China (Hao and Wang, 2005; Han et al., 2011). During our observation period, the median concentrations of SO$_2$, CO, NO$_x$ on haze days were 5.1, 1400 and 27 ppb, respectively. While high, these concentrations are lower than the corresponding concentrations (18, 2200, 75 ppb, respectively) during the extremely severe haze episode that took place in Beijing in January 2013 (Wang et al., 2014b). The median concentration of O$_3$ was 10 ppb on the haze days during our observations, a little bit higher than the severe haze episode in 2013 (<7 ppb; Wang et al., 2014b).

The median levels of SO$_2$, CO, NO$_x$ and O$_3$ were 230%, 50%, 100% and 50% higher, respectively, on the haze days than on the NPF days. SO$_2$, CO and NO$_x$ are usually considered tracers of primary pollution, so their lower levels on the NPF event days indicates that relatively clean conditions favor NPF events (Vahlsing and Smith, 2012;Tian et al., 2018).

### 3.2 Diurnal behavior

In order to draw a clear picture of the evolution of size-segregated particle number concentrations, we analyzed the diurnal concentration behavior of the different trace gases (Figure 5) and particle modes (Figure 6).

Since trace gases have more definitive sources than particles, we can get some insight into particle sources by comparing their diurnal patterns with those of particles in different modes. For instance, CO is usually emitted as the by-product of inefficient

combustion of biomass or fossil fuels (Pétron et al., 2004; Lowry et al., 2016). We
observed similar diurnal patterns for $NO_x$ and CO, with an increase during the morning
rush hours followed by another peak at around 15:00, suggesting similar sources. Due
to lower human activities and traffic during nighttime, lower concentrations of $NO_x$ and
CO were observed. Earlier observations in urban areas having high $NO_x$ concentrations
found that $O_3$ was consumed by its reaction with NO, while $NO_2$ works as precursor
for $O_3$ via photochemical reactions (Wang et al., 2017). In our observations, the diurnal
pattern of $O_3$ was opposite to that of $NO_x$, which is consistent with $O_3$ loss by large
amounts of freshly emitted NO during rush hours and $O_3$ production by photochemical
reactions involving $NO_2$ after the rush hours in the morning.
In Figure 7, we show the median diurnal pattern of particle number size distribution on
the NPF event days and haze days separately. On the NPF event days, we observed
cluster formation from diameters smaller than 3 nm. The growth of newly-formed
particles lasted for several hours, resulting in a consecutive increase of the particle
number concentrations in all the four modes. During traffic rush hours in the morning
and evening, we observed an increase of particle number concentrations in the size
range of cluster mode to around 100 nm.
On the haze days, we still observed an increase of particle number concentration in the
size range of cluster mode to Aitken mode during rush hours. Traditionally, NPF events
occur during the time window between sunrise and sunset by photochemical reactions
(Kerminen et al., 2018). The binary or ternary nucleation between sulfuric acid and
water, ammonia or amines are usually thought of as sources of atmospheric cluster
mode particles, especially in heavily polluted environments (Kulmala et al., 2013;
Kulmala et al., 2014; Yao et al., 2018; Chu et al., 2019). The burst of cluster mode
particle number concentration outside the traditional NPF time window, especially
during the rush hours in the afternoon, suggests a very different source of cluster mode
particles from traditional nucleation, e.g. nucleation from gases emitted by traffic
(Rönkkö et al., 2017).
As shown in Figure 6, on the NPF event days, the cluster mode particle number
concentration started to increase at the time of sunrise and peaked around noon with a
wide single peak, showing the typical behavior related to NPF events (Kulmala et al.,
2012). Comparatively, on the haze days, the cluster mode particle number concentration
showed a double peak pattern similar to the diurnal cycle of $NO_x$ (Figure 5). This
observation in consistent with our discussion above that traffic emission possibly
contributed to cluster mode particles. By comparing cluster mode particle number
concentrations between the haze days and NPF event days, we estimated that traffic-
related cluster mode particles could contribute up to 40-50 % of the total cluster mode
particle number concentration on the NPF event days.
Similar to the cluster mode, the nucleation mode had a single peak on the NPF event

days. Nucleation mode particle number concentrations started to increase shortly after the corresponding increases in the cluster mode, which could be attributed to the growth of cluster mode particles into the nucleation mode. The observed peak of the nucleation mode particle number concentrations had a shoulder at around 7:00 - 9:00 concurrent with the morning peak of the $NO_x$ concentration, which indicates a contribution from traffic to the nucleation mode. It is important to note, however, that the height of this shoulder was only 20% of the maximum nucleation mode particle number concentration. These results suggest that, compared with atmospheric NPF, traffic contributed much less to the nucleation mode particle number concentration.

During the haze days, the diurnal pattern of the nucleation mode particle number concentration reminded that of $NO_x$, showing no peak during the daytime between the rush hours. This suggests that the nucleation mode particles were dominantly from traffic emissions on the haze days. Additionally, it is important to note that during the haze days, we observed different maximum concentrations for morning versus evening peaks, implying a higher contribution of traffic in the morning than in the afternoon. This result is in line with the diurnal cycle of $NO_x$ during the haze days.

On the NPF even days, Aitken mode particles are mainly attributed to two different sources hard to be distinguished from each other: primary and secondary sources, such as combustion and growth of newly formed particles, respectively. In comparison to the cluster and nucleation modes that had pronounced diurnal cycles during the NPF event days, the Aitken mode particle number concentration had a pattern similar to $NO_x$ before 9:00 in the morning. This implies that traffic emissions were important sources to maintain Aitken mode particle concentrations in the morning hours. The Aitken mode particle number concentration increased during the afternoon hours, probably due to the growth of the nucleation mode particles via multicomponent condensation and possibly some other gas-to-particle conversion pathways. The concurrent decrease of the nucleation mode particle number concentration supports this view. The Aitken mode particle number concentration increase in the evening was concurrent with the increase of CO and $NO_x$, which could be attributed to combustion sources (Roberts and Jones, 2004; Koponen et al., 2001).

On the haze days, the Aitken mode particle number concentration experienced little change before about 14:00, contrary to both CO and $NO_x$ concentrations, indicating a small contribution by primary sources during that time of the day. It is important to mention that the growth of particles is not limited to the days when new particle formation occurs. In fact, on the haze days, the wind was typically more stagnant, reducing the vertical mixing of pollutants and their horizontal advection (Zheng et al., 2015). The increase of Aitken mode particle number concentration started at around 16:00 and peaked at around 20:00 similar to the NPF event days. This is concurrent with the increase in the $NO_x$ and CO concentrations, which might be attributed to traffic emissions.

The concentration of accumulation mode particles was an order of magnitude higher during the haze days compared with the NPF days, causing a higher condensation sink (on average 0.015 s$^{-1}$ for the NPF event days and 0.10 s$^{-1}$ for the haze days, as shown in Figure S4), and thus introducing a reason why NPF did not take place on the haze days (Kulmala et al., 2017). The concentration, on the other hand, did not experience much diurnal variation. There was a slight increase in the accumulation mode particle number concentration during the morning rush hours starting at around 6:00 concurrent with the increase in the Aitken mode particle number concentration. The second slight increase started at around 16:00, two hours later than that of the Aitken mode, suggesting a secondary contribution to accumulation mode particles. On the NPF event days, the accumulation mode had the similar diurnal pattern as SO$_2$, implying that SO$_2$ participated in the formation of accumulation mode on the NPF event days.

### 3.3 Correlation between the particle modes and trace gas and PM$_{2.5}$ concentrations

Beijing's atmosphere is a very complicated environment (Kulmala, 2015). Aerosol particles in the atmosphere of Beijing are subject to aerosol dynamical processes, surface reactions, coagulation, deposition and transport, thus hindering direct connection with their sources based on physical size distributions only. However, by correlating each particle mode to various trace gases, we can get indications on the sources of particles. In this section, we use CO, SO$_2$, NO$_x$ and O$_3$ as tracers. By examining responses of size-segregated particle number concentrations to changes in trace gas and PM$_{2.5}$ concentrations (Table 2a and Table 2b), we can get further insights into the main sources of particles in each mode and into the dynamical processes experienced by these particles under different pollution levels. Of course, not all sources or dynamics can be captured using this approach. In addition, due to the complex physical and chemical processes experienced by the particles, the correlation analysis cannot quantify the strength of individual sources or dynamical processes.

### *3.3.1 Connection with SO$_2$*

SO$_2$ is a key precursor for H$_2$SO$_4$ through photochemical reactions in Beijing, which is in turn a requirement for new particle formation in megacity environments (Wang et al., 2013; Yao et al., 2018). Although being a very important precursor of NPF, SO$_2$ had lower concentrations   on the NPF event days than on the haze days (Figure 8). High concentrations of SO$_2$ have been ascribed to regional pollution and anthropogenic condensation sink even in semi-pristine environments (Dada et al., 2017). Earlier observations report that the main sources of SO$_2$ are power plants, traffic and industry, so SO$_2$ can be used as a tracer for regional pollution (Yang et al., 2018;Lu et al., 2010).

Generally, as shown in Figure 8, the SO$_2$ concentration correlated negatively with both

cluster and nucleation mode particle number concentrations. Higher $SO_2$ concentrations were encountered on more polluted days when NPF events were suppressed due to the high particle loadings, explaining the overall negative correlation. However, if we look at the NPF event days and haze days separately, we cannot see any clear correlation between the $SO_2$ concentration and cluster mode or nucleation mode particle number concentration, as shown also in Table 2a and Table 2b. This result indicates that during our observations, NPF occurred in relatively clean conditions, but the strength of a NPF event was not sensitive to the regional pollution level as long as NPF was able to occur.

On the NPF event days, the $SO_2$ concentration correlated positively with the concentrations of both Aitken and accumulation mode particles during the chosen NPF time window, whereas on the haze days no correlation between the $SO_2$ concentration and Aitken mode particle number concentration could be observed. This suggests that regional and transported pollution contributed to Aitken and accumulation mode particles on the NPF event days, while on haze days the transported and regional pollution was only a prominent factor affecting accumulation mode particle number concentration. In addition, $SO_2$ contributes to heterogeneous reactions on particle surfaces, explaining that a fraction of accumulation mode particles could have resulted from the growth of Aitken mode particles (Ravishankara., 1997).

### 3.3.2 Connection with NOx

$NO_x$ is usually considered as the pollution tracer mainly from traffic (Beevers et al., 2012). As shown in Table 2a and Figure 9, the $NO_x$ concentration correlated negatively with both cluster and nucleation mode particle number concentrations on the NPF event days. Compared with the correlation between $SO_2$ and cluster and nucleation mode particle number concentrations, this result indicates that local traffic emissions affected cluster and nucleation mode particles more than regional pollution on the NPF event days.

On the haze days, we did not see any correlation between the cluster mode particle number concentration and $NO_x$ concentration (Table 2b), although according to our analysis above, traffic emissions can be the source of cluster mode particles during the haze days. One possible reason for this is that the relationship between cluster mode particle number concentration and $NO_x$ concentration was not linear. Earlier studies pointed out that the dilution ratio is the dominant factor affecting the number size distribution of nanoparticles generated from traffic gases emissions (Shi and Harrison, 1999; Shi et al., 2001). Temperature and humidity were also identified as factors affecting nanoparticle number size distribution nucleated from tailpipe emissions (Shi et al., 2001). Such factors would decrease the correlation between the cluster and nucleation mode particle number concentrations and $NO_x$ concentration.

The Aitken mode particle number concentration correlated positively with the $NO_x$ concentration on both NPF event days and haze days, suggesting that traffic emissions

might be an important source of Aitken mode particles.
The accumulation mode particle number concentration correlated positively with the
$NO_x$ concentration on the NPF event days, which is consistent with earlier studies
showing that traffic emissions can contribute to accumulation mode particles in urban
areas (Vu et al., 2015). On the haze days, the accumulation mode particle number
concentration correlated less with $NO_x$ than with $SO_2$, suggesting that regional and
transported pollution was a more important contributor to accumulation mode particles
than traffic emissions.

### 3.3.3 Connection with CO

CO has some similar sources as $NO_x$, such as traffic. On the NPF event days, the CO
concentration correlated with particle number concentrations in each mode in a very
similar way as $NO_x$ did, suggesting that CO and $NO_x$ had common sources, such as
traffic emissions, on the NPF event days. This result confirms our analysis above that
traffic emissions could suppress NPF and growth on the NPF event days, in addition to
which they might be important sources of the Aitken and accumulation mode particles.
On the haze days, CO transported from polluted areas dominated the total CO
concentration. The CO concentration had a positive correlation with the accumulation
mode particle number concentration, but no clear correlation with the particle number
concentration of the three other modes. This result confirms our analysis above that on
the haze days, local emissions dominated Aitken particle number concentrations while
regional and transported pollutions affected accumulation mode particle number
concentrations more than local emissions.

### 3.3.4 Connection with $O_3$

Ozone is a secondary pollution trace gas and its concentration represents the oxidization
capacity of atmosphere. Earlier observations found that high $O_3$ concentrations favor
NPF by enhancing photochemical reactions (Qi et al., 2015). However, we did not see
any correlation between the $O_3$ concentration and cluster mode particle number
concentration, suggesting that $O_3$ was not the limiting factor for cluster mode particle
number concentration.
The $O_3$ concentration correlated positively with both nucleation and Aitken mode
particle number concentration on the NPF event days during the NPF time window,
whereas on the haze days $O_3$ concentration correlated only with the Aitken mode
particle number concentration.
The above results suggest that $O_3$ influences heterogeneous reactions and particle
growth rather than the formation of new aerosol particles.

 *3.3.5 Connection to PM2.5*

As shown in Figure 10, the $PM_{2.5}$ concentration correlated negatively with the cluster and nucleation mode particle number concentrations, and positively with the accumulation mode particle number concentration. High $PM_{2.5}$ concentrations tend to suppress NPF by increasing the sinks of vapors responsible for nucleation and growth of cluster and nucleation mode particles. The particles causing high $PM_{2.5}$ concentrations also serve as sinks of cluster and nucleation mode particles by coagulation.

As shown in Table 2a and Figure 12, the Aitken mode particle number concentration correlated positively with the $PM_{2.5}$ concentration on the NPF event days. A possible reason for this could be the tight connection between the Aitken and accumulation mode particles on the NPF event days (Table 3a), and the observation that accumulation mode particles are usually the main contributor to $PM_{2.5}$ in Beijing (Liu et al., 2013). On the haze days, the Aitken mode particle number concentration correlated negatively with the $PM_{2.5}$ concentration (Table 2b). A possible reason for this is that pre-existing large particles acted as a sink for Aitken mode particles by coagulation as well as a sink for vapors responsible for the growth of smaller particles into the Aitken mode. In addition, while $PM_{2.5}$ is dominated by regional and transported secondary aerosols, Aitken mode particles mainly originate from local emissions such as traffic and cooking in Beijing (Wu et al., 2007; Wang et al., 2013; Du et al., 2017; de Jesus et al., 2019).

**3.4 Correlation between different particle modes**

Table 3a and Table 3b as well as Figure 11 show the correlation between particle number concentrations in different modes. On the NPF event days, cluster and nucleation mode particle number concentrations correlated positively with each other due to their common dominant source, NPF. Both cluster and nucleation mode particle number concentrations correlated negatively with the Aitken and accumulation mode particle number concentrations because, as discussed earlier, high concentrations of large particles tend to suppress NPF and subsequent growth of newly-formed particles.

On the NPF event days, Aitken and accumulation mode particle number concentrations correlated positively with each other, as well as with the $SO_2$ and $NO_x$ concentration. This suggests that on the NPF event days, Aitken and accumulation mode particles both formed during regional transportation as secondary particles and were emitted by traffic as primary particles.

On the haze days, cluster and nucleation mode particle number concentrations correlated positively with each other, and with the Aitken mode particle number concentration. This is suggestive of a similar dominating sources for these particle,

most likely traffic emissions. Similar to the NPF event days, cluster and nucleation
mode particle number concentrations correlated negatively with the accumulation mode
particle number concentration, even though this correlation was rather weak (Table 3b).
As expected based on the discussion in section 3.3.5, the Aitken mode particle number
concentration had a negative correlation with the accumulation mode particle number
concentration on the haze days.
**3.5 Atmospheric ions and ion induced nucleation in Beijing**
In order to estimate the contribution of ions to the total cluster mode particle number
concentration and the importance of ion induced nucleation in Beijing, we studied ion
number concentrations in the size range of 0.8-7 nm by dividing them into 3 sub-size
bins: constant pool (0.8-1.5 nm), charged clusters (1.5-3 nm) and larger ions (3-7 nm).
As shown in Figure 12, number concentrations of positive ions were higher than those
negative ions in all the size bins on both NPF event days and haze days. We will only
discuss positive ions here.
The median number concentration of positive ions in the constant pool on NPF event
days was only 100 cm$^{-3}$ in Beijing, much less than that in the boreal forest (600 cm$^{-3}$;
Mazon et al., 2016). Also, the median number concentration of positive charged clusters
was 20 cm$^{-3}$ on the NPF event days, and the ratio to the total cluster mode particle
number concentration was 0.001 to 0.004 during the NPF time window (Figure 13).
This ratio is comparable to that observed in San Pietro Capofiume (0.004), in which the
anthropogenic pollution level was also high, but clearly lower than that observed in
another megacity in China, Nanjing (0.02; Kontkanen et al., 2017). Considerably higher
ratios were observed in clean environments, for example during winter in the boreal
forest at Hyytiälä, Finland (0.7; Kontkanen et al., 2017). The median number
concentration of larger ions (3-7 nm) on the NPF event days was 30 cm$^{-3}$, a little bit
higher than the charged cluster mode particle number concentration, indicating that not
all of the larger ions originate from the growth of charged clusters, but rather from
charging of neutral particles by smaller ions. On the haze days, charged ion number
concentrations were much lower than those on the NPF days, which could be attributed
to the higher condensation sink.
The diurnal pattern of the ratio of number concentration between charged and total
cluster mode particles was the highest during the night with a maximum of 0.008, and
had a trough during daytime with a minimum of 0.001 on the NPF event days. Such
diurnal pattern is similar to earlier observations in Nanjing, San Pietro Capofiume and
Hyytiälä (Kontkanen et al., 2017). This ratio reached its minimum around noon,
because the total cluster mode particle number concentration reached its maximum
around that time due to NPF. The ratio had a small peak at around 9:00, similar to earlier
observations in Centreville and Po Valley (Kontkanen et al., 2016;Kontkanen et al.,

2017). The possible reason is that charged clusters were activated earlier in the morning than neutral clusters. The ratio increased from the midnight until about 4:00, similar to the number concentration of charged clusters.

As shown in Figure 14, the diurnal median of the ratio between the formation rate of positive ions of 1.5 nm ($J_{1.5}^+$) and the total formation rate clusters of 1.5 nm ($J_{1.5}$) varied from 0.0009 to 0.006. This result is comparable to observations in Shanghai, where the positive ion induced nucleation contributed only 0.05% to the total formation rate of 1.7-nm particles ($J_{1.7}$) (Yao et al., 2018).

## 3.6 Particle growth rates

The growth rates of particles generated from NPF events were examined in three size ranges: <3 nm, 3-7 nm and 7- 25 nm (Figure 15). The median growth rates of particles in these size ranges were 1.0 nm/h, 2.7 nm/h and 5.5 nm/h, respectively. The growth rate of cluster mode particles was comparable with that observed in Shanghai (1.5 nm/h; Yao et al., 2018). The notable increase of the particle growth rate with an increasing particle size is a very typical feature in the sub-20 nm size range (Kerminen et al., 2018), and it may also extend to larger particle sizes (Paasonen et al., 2018).

Our observations are in line with the reported range of nucleation mode particle growth rates of 0.1-11.2 nm/h in urban areas of Beijing (Wang et al., 2017b; Jayaratne et al., 2017). Such growth rates can explain the observed increases of Aitken mode particle number concentrations in the afternoon.

## 4    Summary and conclusions

We measured particle number concentrations over a wide range of particle diameters (1.5-1000 nm) on both NPF event days and haze days in winter Beijing. To our knowledge, this was the first time when cluster mode particle number concentrations have been reported on haze days in Beijing.

The observed responses of particle number concentrations in different modes (cluster, nucleation, Aitken and accumulation mode) to changes in trace gas and $PM_{2.5}$ concentrations were quite heterogeneous, suggesting different sources and dynamics experienced by each mode. NPF was the dominant source of cluster and nucleation mode particles. Ion-induced nucleation did not play an important role during the NPF events. The growth rates of cluster and nucleation mode particles increased with an increasing particle size. Traffic emissions contributed to every mode and were the dominant source of cluster and nucleation mode particles on the haze days. The main sources of Aitken mode particles were local emissions, while transported and regional

pollution as well as growth from the nucleation mode also contributed to the Aitken mode. The main source of accumulation mode particles was regional and transported pollution. $PM_{2.5}$ affected the number concentration of sub-100 nm particles by competing for vapors responsible for particle growth and by acting as sinks for particles by coagulation. The main contributors to the $PM_{2.5}$ mass concentration were accumulation mode particles on the haze days.

As demonstrated here and in many other studies (e.g. Brines et al., 2015), ultrafine particles (< 100 nm in diameter) tend to dominate the total aerosol particle number concentration in megacities like Beijing. More attention should therefore put on ultrafine particles in urban environments. We found that both NPF and traffic emissions are important sources of ultrafine particles in Beijing. To improve our understanding on the potential effects of ultrafine particles on health and air quality, we need to do more research on their sources and physical and chemical properties. Laboratory and model analysis on dynamics of ultrafine particles would help us to understand the evolution of particle number size distributions. In addition, to identify and locate other possible sources, long-term observations on ultrafine particles down to the cluster mode as well as source apportionment analyses, such as cluster analysis and receptor model studies, are still needed. Ultrafine particles should also be taken into consideration when making policies to control air pollution. New regulations should be designed to control primary emission sources, such as traffic, or precursor emissions for secondary ultrafine particles involving NPF and subsequent particle growth.

## 5   Acknowledgments

This study received funding from Beijing University of Chemical Technology. This research has received funding from the National Natural Science Foundation of China (41877306). The work is supported by Academy of Finland via Center of Excellence in Atmospheric Science (project no. 272041) and European Research Council via ATM-GTP 266 (742206). LD received funding from the ATM-DP program at university of Helsinki. KRD acknowledges support by the Swiss National Science postdoc mobility grant P2EZP2_181599. LW acknowledges support by National Key R&D Program of China (2017YFC0209505) and the National Natural Science Foundation of China.

*Author contributions.* YZ, YiL, YF, JuK contributed to data collection. YZ, TC, LD contributed to data inversion. YZ and LD contributed to analyzing the data. CY, BC, KRD, FB, TK, YoL, JoK contributed to maintaining the station. YZ, LD, JuK, VMK wrote the paper. TP, LW, JJ, MK provided helpful scientific discussions. All co-authors reviewed the manuscript.

*Competing interests.* The authors declare that they have no conflict of interest.

649 *Data availability:* Particle number concentrations are available upon contacting

650 yingzhouahl@163.com or lubna.dada@helsinki.fi.

651

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

984    Tables and Figures

985    Table 1. Calendar of different types of days during our observations. NPF event days
986    are marked in green and haze days are marked in grey, whereas missing or undefined
987    days are marked in white.

988

989

Table 2a: Correlation coefficients between size segregated particle number concentrations and trace gases mixing ratios/ $PM_{2.5}$ concentration on the NPF event days. The time window was 08:00 - 14:00. High correlation coefficients (|R|>0.5) are marked with bold and italic.

|  | CO | $SO_2$ | $NO_x$ | $O_3$ | $PM_{2.5}$ |
|---|---|---|---|---|---|
| Cluster | -0.61[a] | -0.16[a] | ***-0.66***[a] | 0.16[a] | ***-0.66***[c] |
| Nucleation | -0.5[b] | -0.17[b] | ***-0.55***[b] | 0.36[b] | ***-0.54***[c] |
| Aitken | ***0.58***[b] | ***0.55***[b] | ***0.66***[b] | 0.32[b] | 0.33[c] |
| Accumulation | ***0.71***[b] | ***0.65***[b] | ***0.69***[b] | 0.15[b] | ***0.83***[c] |

[a] included 665 data points (the time resolution was 12 minutes), [b] included 1620 data points (the time resolution was 5 min), [c] included 151 data points (the time resolution was 1 hour).

Table 2b: Correlation coefficients between size segregated particle number concentrations and trace gases mixing ratios/ $PM_{2.5}$ concentration on haze days. The time window was 08:00 - 14:00. High correlation coefficients (|R|>0.5) are marked with bold and italic.

|  | CO | $SO_2$ | $NO_x$ | $O_3$ | $PM_{2.5}$ |
|---|---|---|---|---|---|
| Cluster | -0.19[a] | 0.09[a] | 0.02[a] | 0.13[a] | 0.01[c] |
| Nucleation | -0.24[b] | 0.07[b] | 0.31[b] | 0.17[b] | -0.33[c] |
| Aitken | 0.10[b] | 0.03[b] | 0.44[b] | 0.41[b] | -0.5[c] |
| Accumulation | ***0.71***[b] | ***0.76***[b] | 0.37[b] | 0.17[b] | ***0.81***[c] |

[a] included 620 data points (the time resolution was 12 minutes), [b] included 1460 data points (the time resolution was 5 min), [c] included 89 data points (the time resolution was 1 hour).

Table 3a: Correlation coefficients between particle number concentration of every
mode on NPF event days. The time window was 08:00 - 14:00. High correlation
coefficients (|R|>0.5) are marked with bold and italic.

| | Cluster | Nucleation | Aitken | Accumulation |
|---|---|---|---|---|
| Cluster | 1 | | | |
| Nucleation | 0.76 [a] | 1 | | |
| Aitken | -0.46 [a] | -0.33 [b] | 1 | |
| Accumulation | -0.66 [a] | -0.66 [c] | 0.7 [c] | 1 |

[a] included 516 data points (the time resolution was 12 minutes), [b] included 1251 data
points (the time resolution was 5 min), [c] included 1331 data points (the time
resolution was 5 min).

Table 3b: Correlation coefficients between particle number concentration of every
mode on haze days. The time window was 08:00 - 14:00. High correlation
coefficients (|R|>0.5) are marked with bold and italic.

| | Cluster | Nucleation | Aitken | Accumulation |
|---|---|---|---|---|
| Cluster | 1 | | | |
| Nucleation | 0.74 [a] | 1 | | |
| Aitken | 0.41 [a] | 0.48 [b] | 1 | |
| Accumulation | -0.22 [a] | -0.33 [c] | -0.5 [c] | 1 |

[a] included 342 data points (the time resolution was 12 minutes), [b] included 824 data
points (the time resolution was 5 min), [c] included 845 data points (the time resolution
was 5 min).

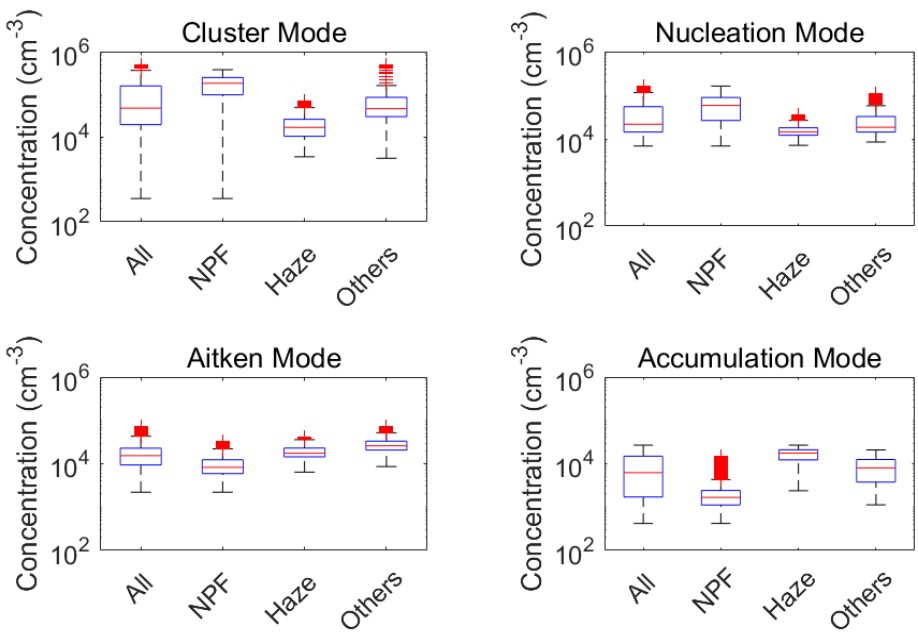


Figure 1. Particle number concentrations in the cluster, nucleation, Aitken and accumulation mode on all the days, NPF event days, haze days and other days. The whiskers include 99.3% of data of every group. Data out of 1.5 × interquartile range are posited outside the whiskers and considered as outliers. The lines in the boxes represent the median value, the lower of the boxes represent 25% of the particle number concentration and the upper of the boxes represent 75% of the particle number concentration. Data marked with red pluses represent outliers.


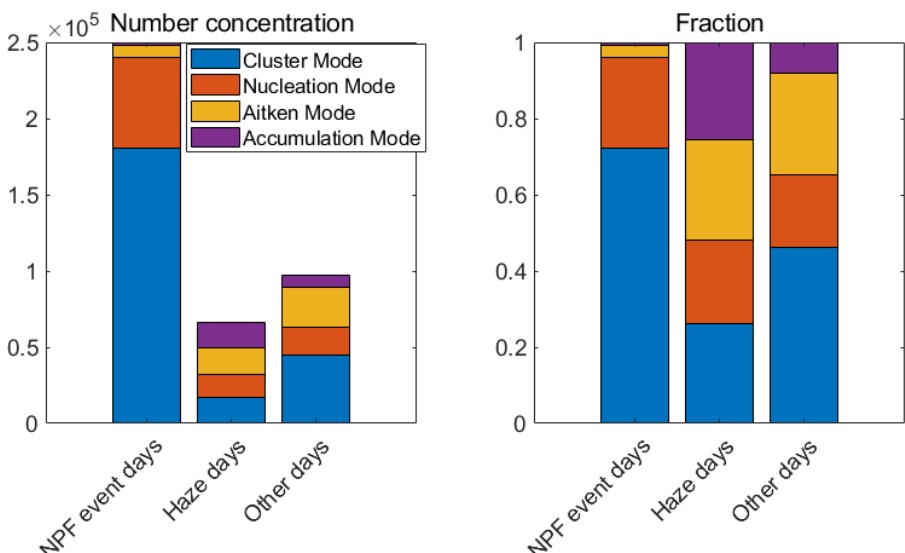


Figure 2. The median size-segregated number concentrations (left) and the median fraction of each mode to the total particle number concentration (right) on the NPF event days, haze days and other days.


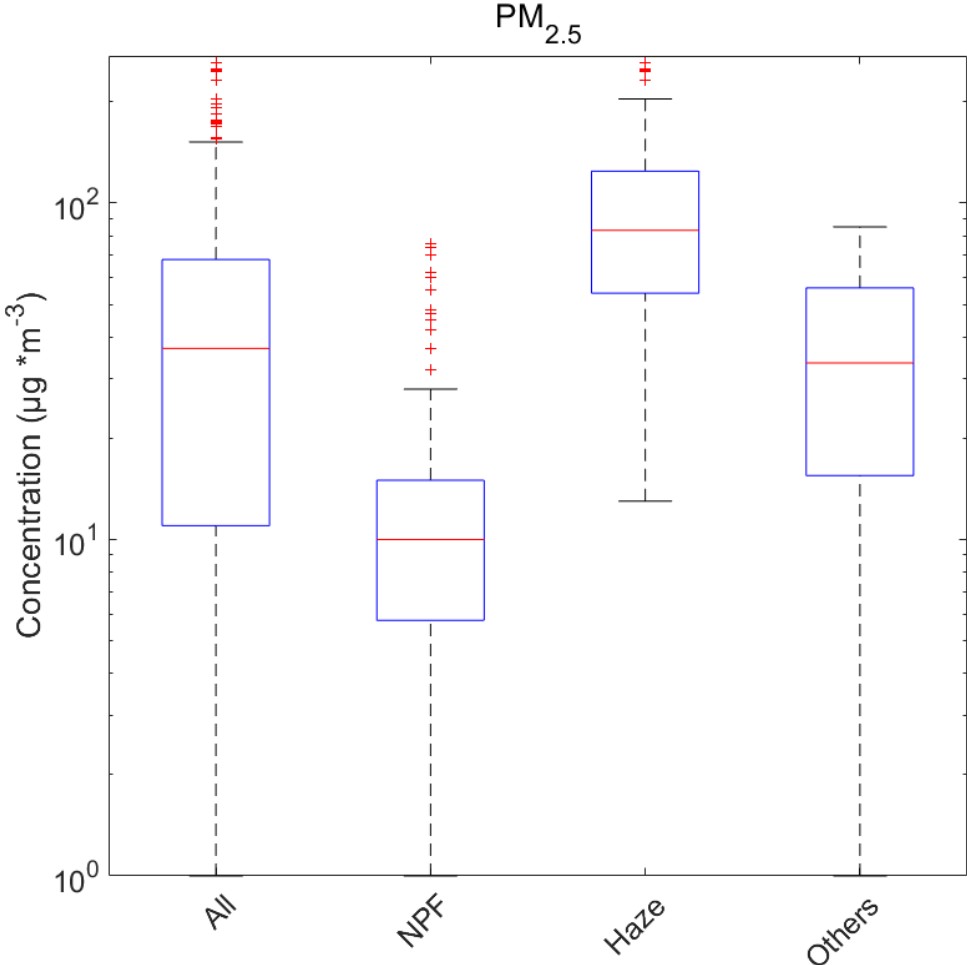


Figure 3. General character of the PM$_{2.5}$ mass concentration on all the days, NPF event days, haze days, and others days. The boxes show the median (red line) and 25% and 75% percentiles of the PM$_{2.5}$ mass concentration. Data marked with red pluses represent outliers as in Figure 1.

1040

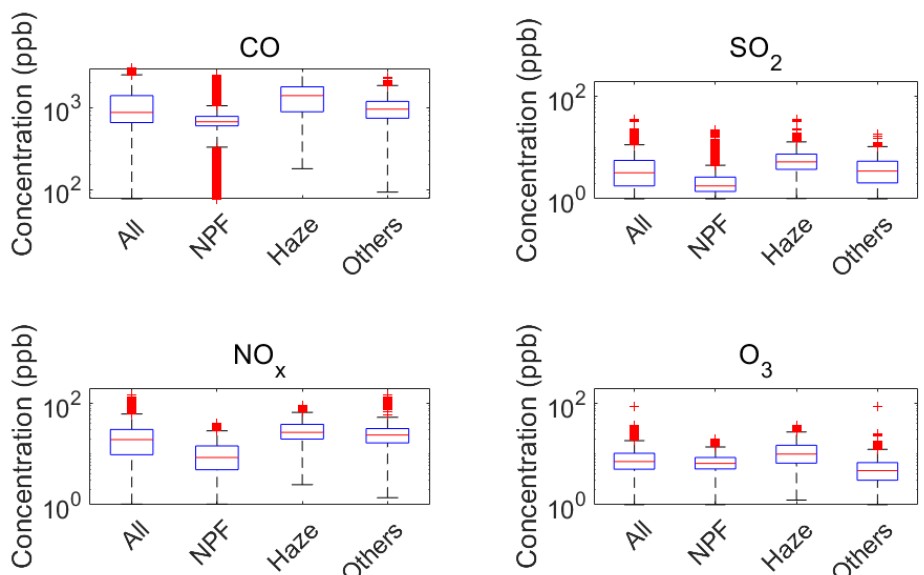

1041

Figure 4. Trace gases mixing ratios of CO, $SO_2$, $NO_x$ and $O_3$ on all the days, NPF event days, haze days and other days. The boxes show the median (red line) and 25% and 75% percentiles of the mixing ratios. Data marked with red pluses represent outliers as in Figure 1.



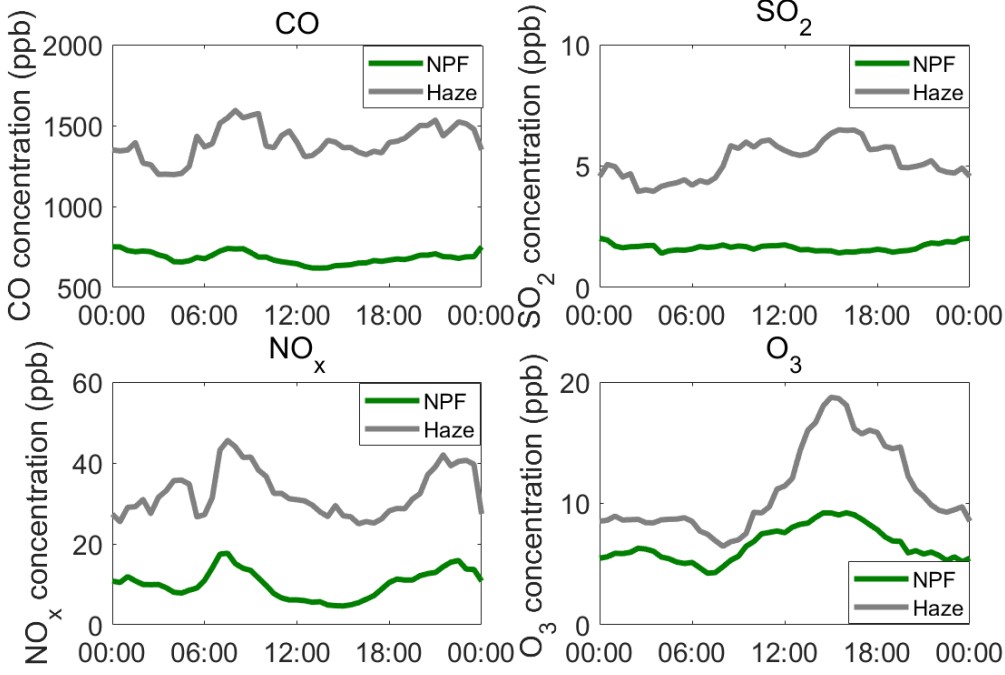


Figure 5. Diurnal variation of trace gas (CO, SO$_2$, NO$_x$ and O$_3$ separately) mixing
ratios on the NPF event days (green lines) and haze days (grey lines) separately. The
time resolution was 30 minutes for every data point. Every data point here represents
the median of all data at the same time of the days.



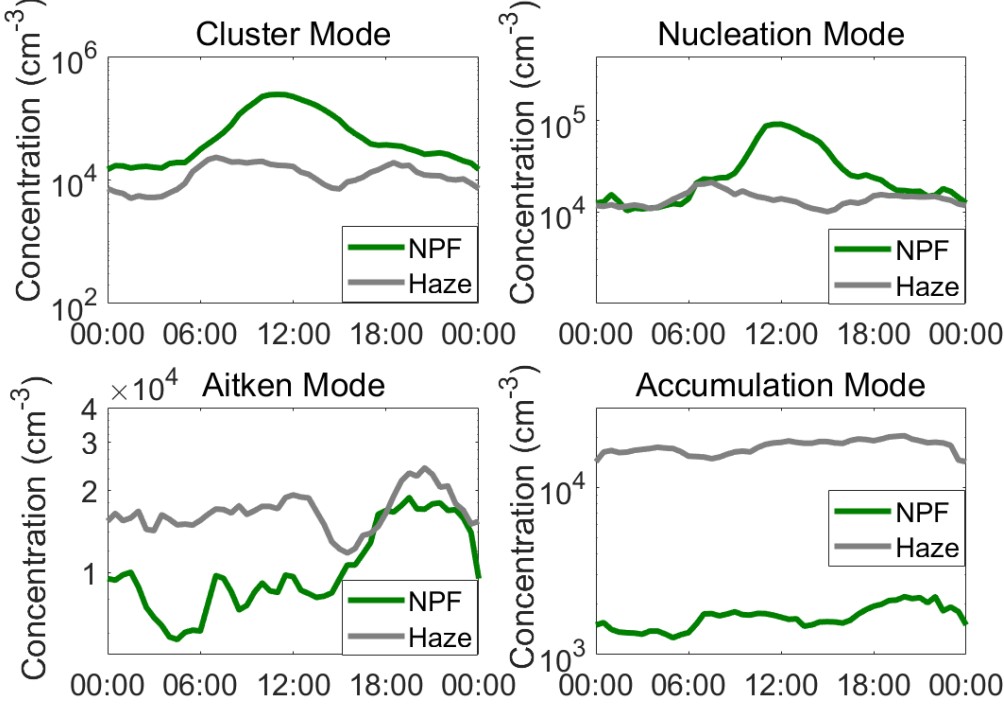


Figure 6. Diurnal variation of particle number concentration of every mode (cluster,
nucleation, Aitken and accumulation mode separately) on the NPF event days (green
lines) and haze days (grey lines). The time resolution was 30 min for every data point.
Every data point here represents the median of all data at the same time of the days.

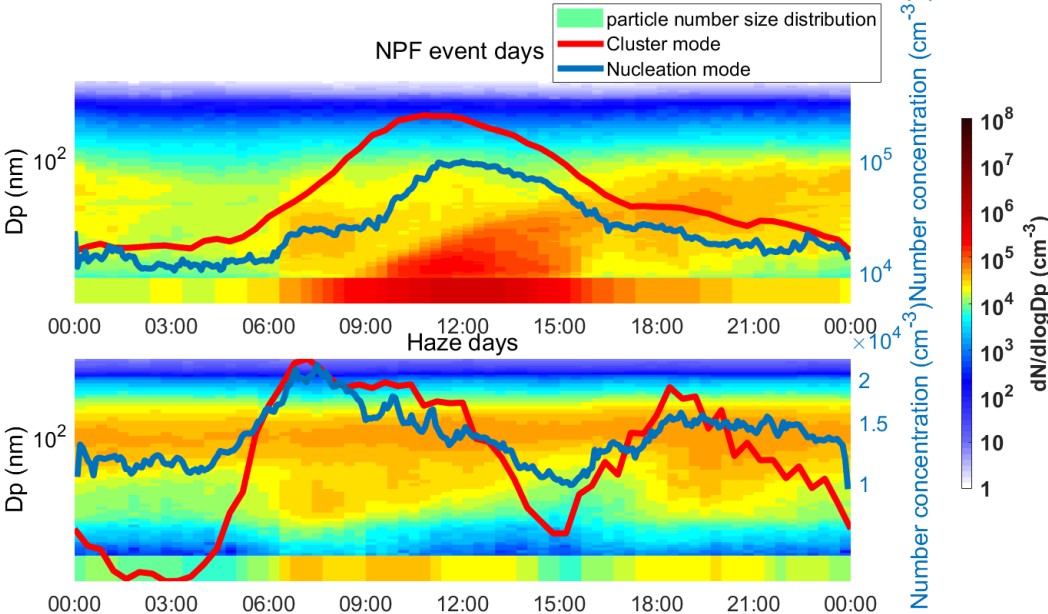

Figure 7. Median diurnal patterns of the particle number size distribution over the size range of 1.5-1000 nm and number concentrations of cluster mode (red lines) and nucleation mode (blue lines) particles on the NPF event days (upper panel) and haze days (lower panel). The time resolution for every data point of particle number size distribution and cluster mode particle number concentration was 12 minutes. The time resolution of every data point of nucleation mode particle number concentration was 5 minutes.


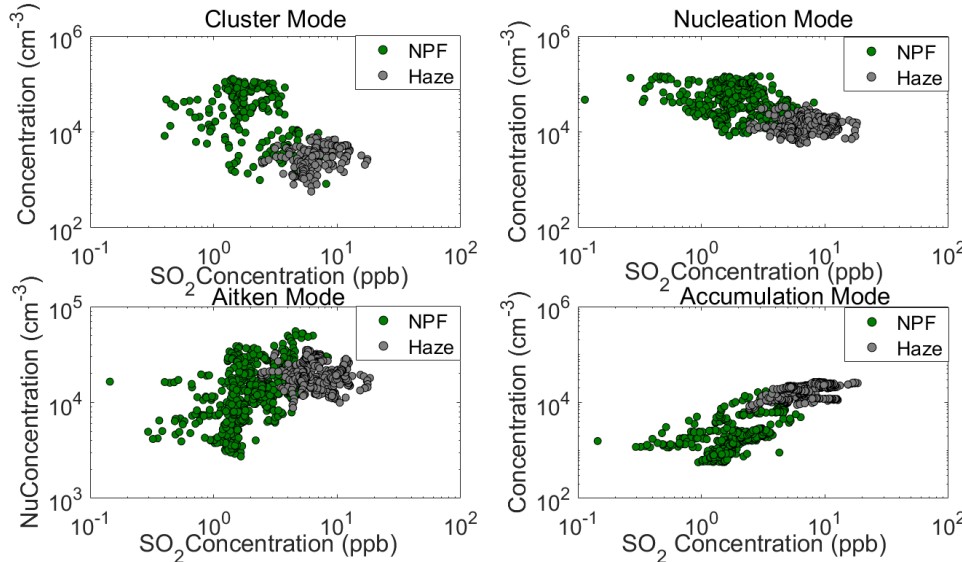


Figure 8. Relation between the SO₂ concentration and particle number concentration in
each mode. The time resolution of the data points was 1 hour.


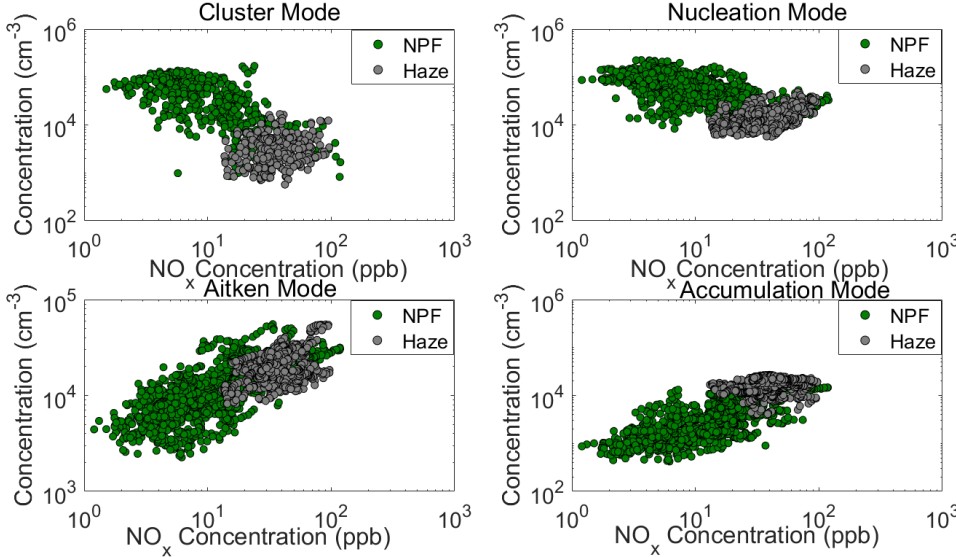


Figure 9. Relation between the $NO_x$ concentration and particle number concentration
in each mode. The time resolution of the data points was 1 hour.

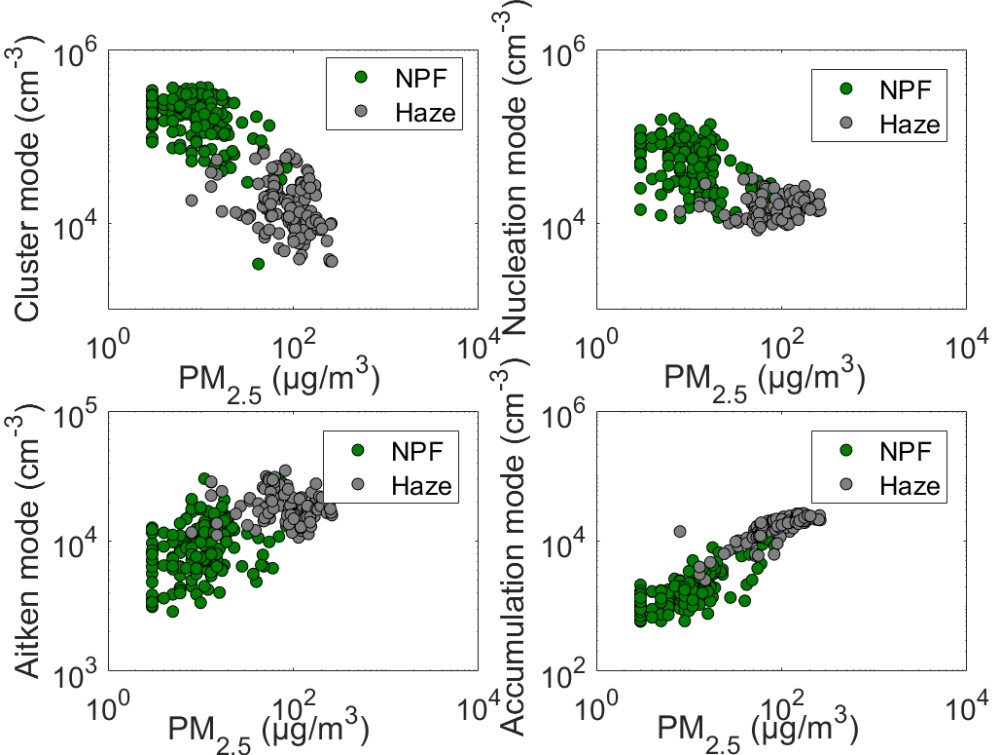


Figure 10. Correlation between $PM_{2.5}$ concentration and particle number concentration in each mode on the NPF event days (green dots) and haze days (grey dots) separately. The time resolution of the data points was 1 hour.



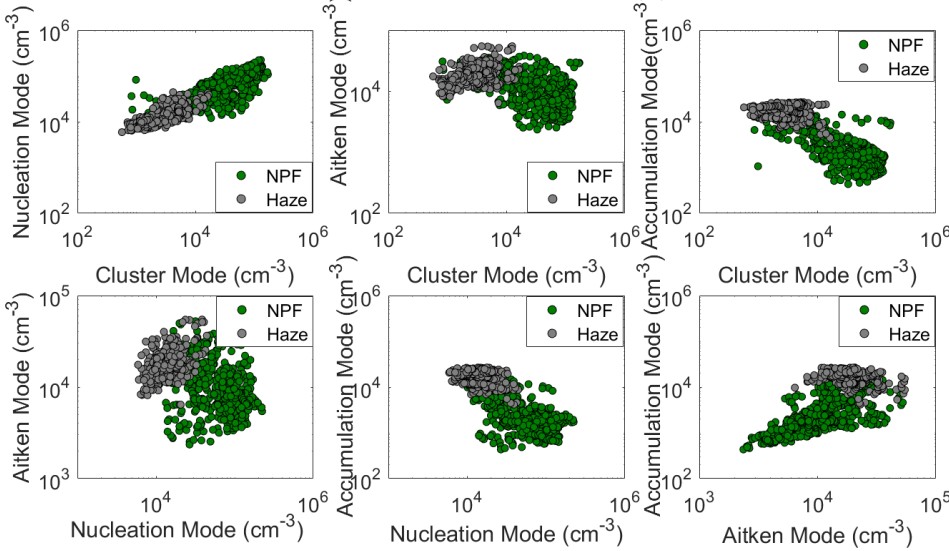


Figure 11. Correlation between every mode each other on NPF event days (green
dots) and haze days (grey dots). The time resolution of data in the plots of correlation
between cluster mode and other modes was 12 min and the time resolution of other
data points was 5 min.

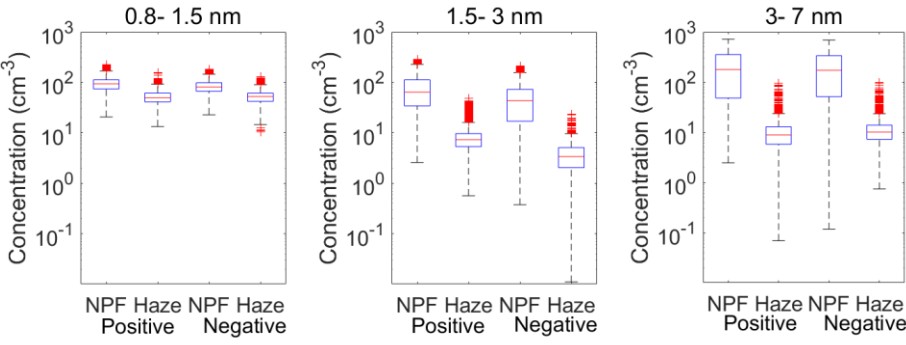


Figure 12. Positive and negative ion number concentrations in the size bins of 0.8-
1.5nm, 1.5-3 nm and 3-7 nm on NPF event days and haze days separately. The whiskers
include 99.3% of data of every group. Data out of 1.5 × interquartile range are posited
outside the whiskers and considered as outliers. The lines in the boxes represent the
median value, the lower of the boxes represent 25% of the number concentration, and
the upper of the boxes represent 75% of the number concentration. Data marked with
red pluses represent outliers.


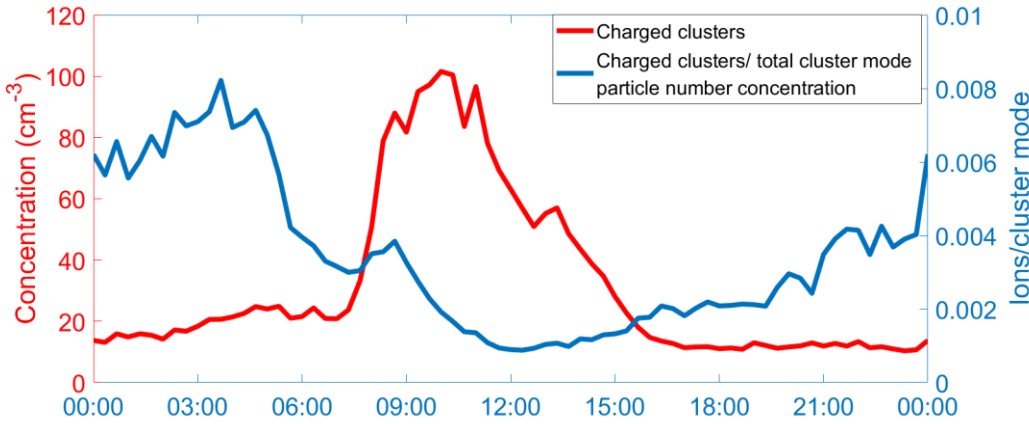


Figure 13. Diurnal pattern of charged clusters (1.5-3 nm) number concentration (red
line) and ratio of charged clusters to total cluster mode (1.5-3 nm) particle number
concentration on the NPF event days (blue line). The time resolution of the used data
was 12 min.


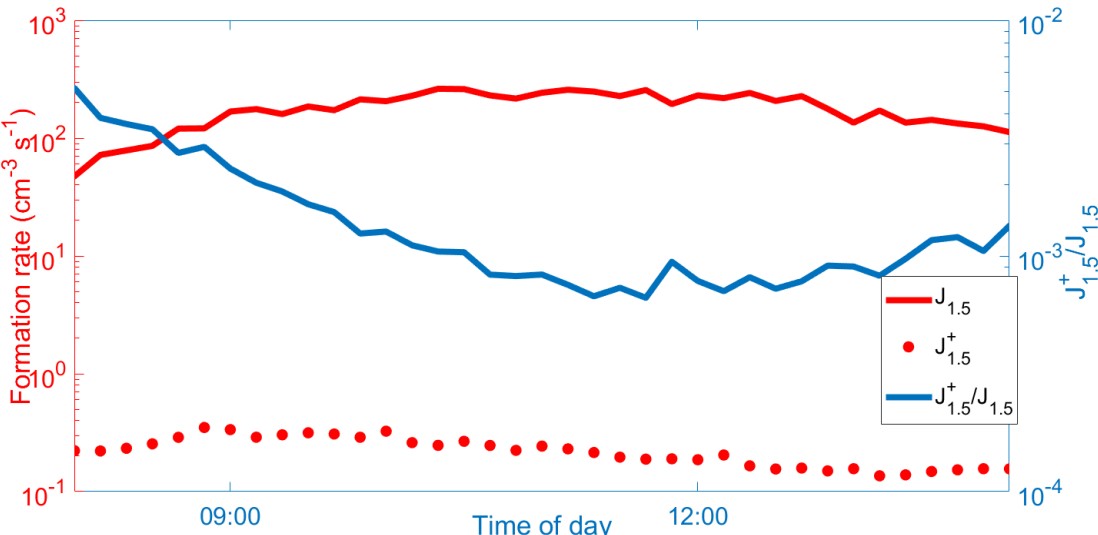


Figure 14. Diurnal pattern of formation rate of positive charged clusters of 1.5 nm (red dots) and neutral clusters of 1.5 nm (red line) and the ratio between them (blue line) on the NPF event days during the NPF time window we chose. The time resolution of the used data was 12 min.




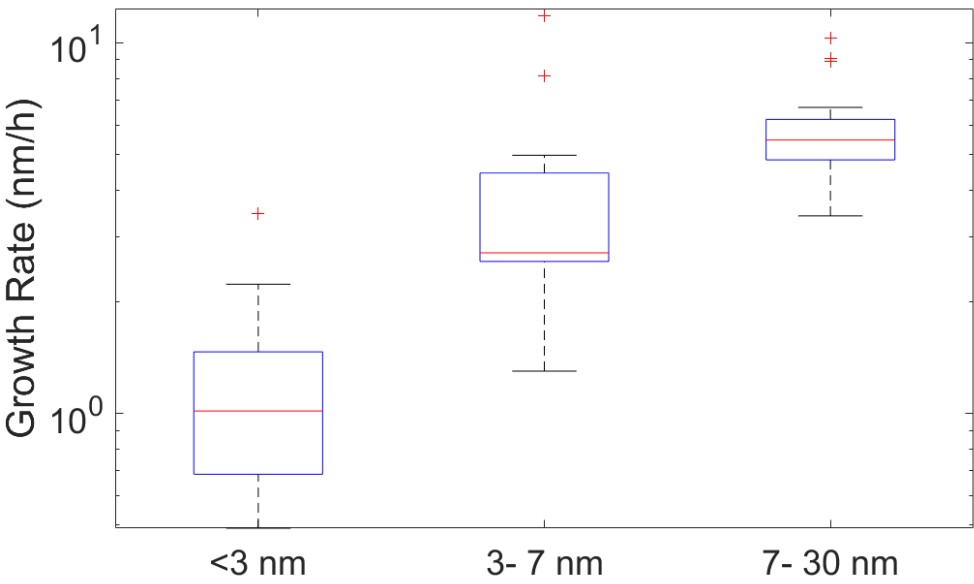


Figure 15. Growth rates of cluster mode and nucleation mode particles generated from NPF events. The lines in the boxes represent the median value, the lower of the boxes represent 25% of the growth rates and the upper of the boxes represent 75% of the growth rates. Data marked with red pluses represent outliers.

