# Peer review of "Variation of size-segregated particle number concentrations in winter"

_Atmospheric Chemistry and Physics, 2019_

## Referee Comment (RC1) · Anonymous Referee #1 · 13 May 2019

This manuscript analyzed 3 months continuous measurement of particle size distribution from 1.2 nm to 10000 nm during winter 2018 in Beijing. This kind of observation, that cover almost the full range of particle size and include both charged and neutral clusters/particles, is rather limited in China. New particle formation and haze days were discussed separately, and found a clear correlation between the cluster and nucleation modes during NPF days. In addition, the work found that all modes in the sub-micron size range were correlated with NOx, indicate traffic emission can contribute to all particle sizes. In general, the manuscript did provide useful information and knowledge, but some more in-depth analysis is encouraged. The manuscript is in general well written and documented. The topic fits well in the scope of ACP. I recommend this manuscript can be published after some revisions.

[Figure]

Comments: 1. More discussions on the charged ions/clusters from NAIS are encouraged. Can the ion induced nucleation be observed? Is it important?

2. There are some overlap for the particle size distribution between NAIS and PSD. It would be good that the authors can provide some information about the intercomparison between these two techniques.

3. I would suggest to provide 1 plot to show the traffic emission derived increase of cluster and nucleation mode particles, and maybe the correlation plot between cluster mode particles and NOx during the non-NPF days.

4. It is a bit unusual that there were no overlap between NPF and haze days. There were quite many studies that observed NPF with considerable high concentrations of PM2.5. What will happen if classify the haze days by the concentration of PM2.5, i.e. 100 ug/m3?

5. Table 2, change the color-marked numbers to, i.e. bold or italic.

6. Figs. 7-9, where were the data points of "others"?

7. Fig. 9, there showed pretty good correlation between Aitken mode particles and accumulation mode particles during NPF days. What's the possible reason?
* * *

---

## Referee Comment (RC2) · Anonymous Referee #2 · 29 Aug 2019

**Comments to ACP-2019-60**

**General comments**

The manuscript describes the evolution of aerosol size-segregated particle number concentration during winter 2018 in Beijing. The data is separated in two different sets: days when haze is observed, and days with new particle formation (NPF) events. Additionally, the particle size distribution is separated into different modes according to the particle diameter: cluster, nucleation, Aitken and accumulation modes. Trace gases concentrations are used to establish the origin of the aerosol observed and induce a primary or secondary origin of the particles observed in each mode.

The topic of this paper is within the scope of this journal and the dataset used is interesting because particles in a wide size range, including very small particles (1.5 to 1000 nm) are measured. However, I believe that with such an interesting dataset one could expect a more comprehensive study. For example, it would be nice to see the evolution of condensation sinks during NPF and haze days, or calculate growth rates for each mode during NPF days. Additionally, the statistical data analysis is very simplistic, and the authors should reconsider analyzing the data with a different approach. The only statistical tools used for relating trace gases with the different modes are correlation coefficients. Assuming that the processes involved in the formation of the different modes is linear is a big oversimplification. The authors should improve the data analysis or discuss the limitations of the methodology they use.

In general, the manuscript is poorly written. There are grammar mistakes and the language is not fluent. This makes it difficult to follow some parts of the manuscript. The authors should have the paper proof-read and edited by a competent English speaker before publishing it.

Most of the discussions are very short and do not provide much more information than what is presented in the tables or figures.

The figures are good in general. The scales on figures 5 and 6 could be improved, and there are a few technical mistakes and information missing in the figure captions (see comments below).

**Specific comments**

Table 1: It would be useful for the reader if the authors mention in the text the number of days classified as NPF and haze.

I don't see the definition of the modes. Also, which instrumentation did you use to calculate the different modes number concentrations? There are overlapping size ranges for nucleation and Aitken modes in the instrumentation you described.

Section 2.2: Please include the time resolution of the data you use. I could not find this information for the PSD system and trace gases. How did you merge the data when the different instruments have different time resolutions? It is important that you describe this procedure carefully.

Line 181: "In general, there were no overlap between NPF and haze periods". Did you look for NPF events during haze days? If haze and NPF are not 100% mutually excluding it would be interesting to describe these episodes. If they are, then change your sentence to make it clear that there was never an overlap. Also, did you determine haze days or did the China Meteorological Administration do this? If the authors did the classification, they should include the instrumentation used.

Lines 222-228: The authors should also talk about $O_3$ here. I only see information for $SO_2$, CO and NOx.

Lines 232-233: NPF does not favor clean environments. In any case, clean environments favor NPF.

Lines 247-251: "NOx and CO are important precursors of $O_3$ in Chinese urban areas. Based on our data, $O_3$, on the other hand, started to increase […] after the levels of NOx and CO started to decrease". Your wording is confusing. It feels like you are suggesting that NOx and CO are not precursors of $O_3$. Please reword.

Lines 280-284: I don't see Aitken mode concentrations being similar to NOx evolutions before 9:00. See my comment on Figure 6 below.

Lines 295-297: Did you measure meteorological parameters or is this a general statement? If it is a general statement change "the wind was" for "the wind is…".

Line 301: It would be interesting to see the graphs for CS instead of giving only a daily value.

Section 3.3: In line 318 the authors state "In this section, we use CO, SO2, NOx and O3 as tracers", but there are no comments whatsoever regarding CO or O3 in this section.

Line 337: Looking at the figure, it doesn't seem to me that SO2 and cluster and nucleation mode concentrations are correlated, especially for NPF days. What are the correlation coefficients for NPF days and haze days separately? (see also comment for Table 2).

Lines 400-402: Please elaborate and comment on the correlation with the other modes. PM2.5 is also highly correlated with cluster mode but this is not discussed. Consider showing the correlations in an additional figure.

Conclusion: This section is written as a summary. The conclusion should reflect the significance of the results presented in this paper compared with existing observations, and give a message beyond summarizing what has already been said in the previous sections.

Line 415-416: I do not see in the text where this is discussed (secondary sources contribution to the Aitken mode during haze days).

Table 2: I think this table would be more useful if the authors separate the data for haze days and NPF days. Also, what do you mean by "all the data are in log scale"? Please reword. You are showing correlation coefficients here, which are not represented in any scale.

Figures 1, 2, 3, 4: Are these daily averages? Please specify the data you used to make the plot.

Figures 5 and 6: Please specify the time resolution of the data you are showing, and reword "and they are the median data from midnight to midnight".

Figure 5: The scale in the upper left graph is different to the others. If you decide to use the same scale, change it to match the others. If it is not important for you that the graphs have the same scale, change the other scales (especially SO2 and O3) so that the variations can be seen more clearly.

Figure 6: The scales used here do not allow to see changes in the Aitken and accumulation modes. I would suggest changing the scales on the lower graphs. It is hard to see the changes you mention in the discussion.

**Technical comments**

Line 93-94: "…. complicating the story even further". I would suggest using a different language.

Line 153: Please change the verb tense: measures -> measured.

Lines 169 and 175: Correct the references format.

Line 213-214: Check the Aitken and accumulation median concentrations. Are they exactly the same?

Line 300: Change "maybe" for "may be".

Line 305: Delete "in" between "increase" and "during".

Line 311: Add in: "… SO2 participated in the formation…"

Line 386: "resulting in an increase…"

Figure 2: Switch "left" and "right" in the figure caption.

Figure 3: Please change the label: OtherS -> Others.

---

## Author Comment (AC1) · 11 Nov 2019

This manuscript analyzed 3 months continuous measurement of particle size distribution from 1.2 nm to 10000 nm during winter 2018 in Beijing. This kind of observation, that cover almost the full range of particle size and include both charged and neutral clusters/particles, is rather limited in China. New particle formation and haze days were discussed separately, and found a clear correlation between the cluster and nucleation modes during NPF days. In addition, the work found that all modes in the sub-micron size range were correlated with NOx, indicate traffic emission can contribute to all particle sizes. In general, the manuscript did provide useful information and knowledge, but some more in-depth analysis is encouraged. The manuscript is in general well written and documented. The topic fits well in the scope of ACP. I recommend this manuscript can be published after some revisions.

We would like to thank the referee for the suggestions and careful editorial comments. Our replies (text in blue) to the comments (text in black) item by item and modification in our manuscript (text in red) as per suggestions of the referee are presented as below:

Comments:

More discussions on the charged ions/clusters from NAIS are encouraged. Can the ion induced nucleation be observed? Is it important?

We thank the reviewer for their suggestions. We did observe ion induced nucleation during our observation however we think that it constitutes only a minor fraction in comparison to neutral nucleation mechanism. Both cluster ion number concentration and the formation rate of 1.5 nm ions constituted a small fraction of total clusters number concentration and total formation rate of 1.5 nm clusters.

The following discussion and figures (Figure R1-1, Figure R1-2, Figure R1-3 and Figure R1-4) were added to the manuscript:

2.4.1 Calculation of the growth rate

[revised manuscript text omitted]

There are some overlap for the particle size distribution between NAIS and PSD. It would be good that the authors can provide some information about the inter comparison between these two techniques.

Indeed, we added the discussion on instrument comparison and related figures (Figure R1-5 and Figure R1-6) to the manuscript (line 186) as per suggestion of the referee.

The Particle Size Distribution system (PSD) and Neutral Cluster and Air Ion Spectrometer (NAIS) had an overlapping particle size distribution over the mobility diameter range of 3-42 nm. As shown in Figure R1-5, total particle number concentrations from the NAIS and PSD system correlated well with each other on both NPF event days ($R^2$ was 0.92) and haze days ($R^2$ was 0.90) in the overlapping size range. The slopes between the total particle number concentration from the PSD system and that from the NAIS were 0.90 and 0.85 on the NPF event days and haze days, respectively. The particle number size distribution in the overlapping size range of the NAIS and PSD system matched well on both NPF event days and haze days as shown in Figure R1-6.

I would suggest to provide 1 plot to show the traffic emission derived increase of cluster and nucleation mode particles, and maybe the correlation plot between cluster mode particles and $NO_x$ during the non-NPF days.

We thank the referee for the suggestions. We added Figure R1-7 to the manuscript. Correspondingly, we updated our discussion in lines 252- 259 as below:

In Figure R1-7, we show the median diurnal pattern of particle number size distribution on the NPF event days and haze days separately. On the NPF event days, we observed cluster formation from diameters smaller than 3 nm. The growth of newly-formed particles lasted for

several hours, resulting in a consecutive increase of the particle number concentrations in all the four modes. During traffic rush hours in the morning and evening, we observed an increase of particle number concentrations in the size range of cluster mode to around 100 nm.

On the haze days, we still observed an increase of particle number concentration in the size range of cluster mode to Aitken mode during rush hours. Traditionally, NPF events occur during the time window between sunrise and sunset by photochemical reactions (Kerminen et al., 2018). The binary or ternary nucleation between sulfuric acid and water, ammonia or amines are usually thought of as sources of atmospheric cluster mode particles, especially in heavily polluted environments (Kulmala et al., 2013;Kulmala et al., 2014; Yao et al., 2018; Chu et al., 2019). The burst of cluster mode particle number concentration outside the traditional NPF time window, especially during the rush hours in the afternoon, suggests a very different source of cluster mode particles from traditional nucleation, e.g. nucleation from gases emitted by traffic (Rönkkö et al., 2017).

As shown in Figure 6, on the NPF event days, the cluster mode particle number concentration started to increase at the time of sunrise and peaked around noon with a wide single peak, showing the typical behavior related to NPF events (Kulmala et al., 2012). Comparatively, on the haze days, the cluster mode particle number concentration showed a double peak pattern similar to the diurnal cycle of $NO_x$ (Figure 5). This observation in consistent with our discussion above that traffic emission possibly contributed to cluster mode particles. By comparing cluster mode particle number concentrations between the haze days and NPF event days, we estimated that traffic-related cluster mode particles could contribute up to 40-50 % of the total cluster mode particle number concentration on the NPF event days.

It is a bit unusual that there were no overlap between NPF and haze days. There were quite many studies that observed NPF with considerable high concentrations of $PM_{2.5}$. What will happen if classify the haze days by the concentration of $PM_{2.5}$, i.e.100 ug/m$^3$?

We thank the referee for the suggestions. We observed NPF events and haze events on the same days, but not at the same time. We classified haze days not only according to the visibility and relative humidity but also according to the time period haze events lasted. Days were classified as haze days when haze lasted for at least 12 consecutive hours. According to this classification, we did not observe any overlap between NPF event days and haze days.

As per suggestion to the referee, we classified haze days by the concentration of $PM_{2.5}$, i.e.100 μg/m$^3$ . We show time series of particle number size distribution and $PM_{2.5}$ concentration during our observations in Figure R1-8. In Figure R1-8, the NPF events can be identified with the 'banana shapes'. We did not observe any NPF events happening at the same time when $PM_{2.5}$ concentration was higher than 100 μg/m$^3$.

Table 2, change the color-marked numbers to, i.e. bold or italic.

We modified Table 2 as per suggestion to the referee.

Figs. 7-9, where were the data points of "others"?

The data points of 'others' are those that do not belong to the classification of NPF or haze days. They are usually polluted days during which we do not observe any NPF, but cannot be

classified as haze days due to low PM loading resulting in not so bad visibility. To make the plots clearer, we only present NPF and haze days here.

Fig. 9, there showed pretty good correlation between Aitken mode particles and accumulation mode particles during NPF days. What's the possible reason?

On the NPF event days, Aitken and accumulation mode particle number concentrations correlated positively with each other, as well as with the $SO_2$ and $NO_x$ concentration. This suggests that on the NPF event days, Aitken and accumulation mode particles were both formed during regional transportation as secondary particles and were emitted by traffic as primary particles.

To make the discussion on correlation between each of the modes, we updated the discussion at section 3.4 and separate Table 3 into Table R1a and Table R1b according NPF event days and haze days, in addition we changed Figure 9 into Figure R1-9 as following:

3.4 Correlation between different particle modes

[revised manuscript text omitted]

Figure R1-4: An example of how the appearance time method was used to calculate growth rate. The appearance time was recorded as a function of particle diameter as the black stars in the figure. The black lines are the fitted growth periods. The growth rates were calculated by calculating the slopes of the black lines.

[Figure]

Figure R1-5: Total particle number concentration in size range of 3-42 nm from NAIS and PSD system. There are 1271 data points on the plots of NPF days and 887 data points on the plots of haze days. The time resolution was 5 minutes.

[Figure]

Figure R1-6: Median particle number size distribution of data from NAIS (blue line) and PSD system (red line) on NPF event days (left panel) and haze days (right panel) during our observation. The time resolution we used here for every point was 1h.

[Figure]

Figure R1-7: Median diurnal patterns of the particle number size distribution over the size range of 1.5-1000 nm and number concentrations of cluster mode (red lines) and nucleation mode (blue lines) particles on the NPF event days (upper panel) and haze days (lower panel). The time resolution for every data point of particle number size distribution and cluster mode particle number concentration was 12 minutes. The time resolution of every data point of nucleation mode particle number concentration was 5 minutes.

[Figure]

Figure R1-8: Time series of particle number size distribution and PM$_{2.5}$ concentration (blue line) during our observation period.

[Figure]

Figure R1-9: Correlation between every mode each other on NPF event days (green dots) and haze days (grey dots). The time resolution of data in the plots of correlation between cluster mode and other modes was 12 min and the time resolution of other data points was 5 min.

---

## Author Comment (AC2) · 11 Nov 2019

The manuscript describes the evolution of aerosol size-segregated particle number concentration during winter 2018 in Beijing. The data is separated in two different sets: days when haze is observed, and days with new particle formation (NPF) events. Additionally, the particle size distribution is separated into different modes according to the particle diameter: cluster, nucleation, Aitken and accumulation modes. Trace gases concentrations are used to establish the origin of the aerosol observed and induce a primary or secondary origin of the particles observed in each mode.

The topic of this paper is within the scope of this journal and the dataset used is interesting because particles in a wide size range, including very small particles (1.5 to 1000 nm) are measured. However, I believe that with such an interesting dataset one could expect a more comprehensive study. For example, it would be nice to see the evolution of condensation sinks during NPF and haze days, or calculate growth rates for each mode during NPF days. Additionally, the statistical data analysis is very simplistic, and the authors should reconsider analyzing the data with a different approach. The only statistical tools used for relating trace gases with the different modes are correlation coefficients. Assuming that the processes involved in the formation of the different modes is linear is a big oversimplification. The authors should improve the data analysis or discuss the limitations of the methodology they use.

In general, the manuscript is poorly written. There are grammar mistakes and the language is not fluent. This makes it difficult to follow some parts of the manuscript. The authors should have the paper proof-read and edited by a competent English speaker before publishing it.

Most of the discussions are very short and do not provide much more information than what is presented in the tables or figures.

The figures are good in general. The scales on figures 5 and 6 could be improved, and there are a few technical mistakes and information missing in the figure captions (see comments below).

We would like to thank the referee for the suggestions and careful editorial comments. These comments are valuable and very helpful for revising and improving our paper. We have studied all the comments carefully and made corrections. We would like to thank the referee for the suggestions and careful editorial comments. Our replies (text in blue) to the comments (text in black) item by item and modification in our manuscript (text in red) as per suggestions of the referee are presented as below:

As suggested by both referees, we improved the scope of this manuscript by adding discussions on atmospheric ions in size range of 0.8-7 nm and ion induced nucleation, these discussions improve our understanding of the sources of cluster mode particles. Also, we made a more comprehensive study about traffic- related cluster and nucleation mode particles as suggested by both referees. In addition, we added results on growth rates on both NPF event days of cluster and nucleation mode particles as suggested by the referee.

We agree that correlation coefficients analysis cannot tell much on the evolution of size-segregated particle number concentration accurately because the processes involved in the formation of different modes should not be linear as pointed out by the referee. By examining responses of size-segregated particle number concentrations to changes in trace gas and $PM_{2.5}$ concentrations (Table R2a and Table R2b), we can get further insights into the main sources of

particles in each mode and into the dynamical processes experienced by these particles under different pollution levels. Of course, not all sources or dynamics can be captured using this approach. In addition, due to the complex physical and chemical processes experienced by the particles, the correlation analysis cannot quantify the strength of individual sources or dynamical processes.

According to the reviewer's good instruction, we updated some parts of our manuscript on both languages and scientific discussion as shown in the replies to specific comments.

As per suggestions of both referees, the following discussion on parameter calculations and Figure R2-1was added to our manuscript as section 2.4.

2.4 Parameter calculation

2.4.1 Calculation of the growth rate

The growth rates of cluster and nucleation mode particles were calculated from positive ion data and particle data from Neutral Cluster and Air Ion Spectrometer (NAIS), respectively, by using the appearance time method introduced by Lehtipalo et al. (2014). In this method, the particle number concentration of particles of size $dp$ is recorded as a function of time, and the appearance time of particles of size $dp$ is determined as the time when their number concentration reaches 50% of its maximum value during new particle formation (NPF) events.

The growth rates (GR) were calculated according to:

$$GR = \frac{dp_2 - dp_1}{t_2 - t_1} \tag{1}$$

where $t_2$ and $t_1$ are the appearance times of particles with sizes of $dp_2$ and $dp_1$ respectively. Figure R2-1 shows an example of how this method was used.

2.4.2 Calculation of the coagulation sink

The coagulation sink (CoagS) was calculated according to the equation (2) introduced by Kulmala et al. (2012):

$$CoagS_{dp} = \int K(dp, d'p)n(d'p)dd'p \cong \sum_{d'p=dp}^{d'p=max} K(dp, d'p)N_{d'p} \tag{2}$$

where $K(dp, d'p)$ is the coagulation coefficient of particles with sizes of $dp$ and $d'p$, $N_{d'p}$ is the particle number concentration with size of $d'p$.

2.4.3 Calculation of the formation rate

The formation rate of 1.5-nm particles ($J_{1.5}$) was calculated using particle number concentrations measured with a Particle Sizer Magnifier (PSM). The formation rate of 1.5-nm ions ($J_{1.5}^{\pm}$) was calculated using positive and negative ions data from the Neutral Cluster and Air Ion Spectrometer (NAIS) as well as PSM data. The upper limit used was 3 nm. The values

of $J_{1.5}$ and $J_{1.5}^{\pm}$ were calculated following the methods introduced by Kulmala et al. (2012) with equation (3) and equation (4), respectively:

$$J_{dp} = \frac{dN_{dp}}{dt} + CoagS_{dp} \cdot N_{dp} + \frac{GR}{\Delta dp} \cdot N_{dp} \qquad (3)$$

where $CoagS_{dp}$ is coagulation sink in the size range of $[dp, dp + \Delta dp]$ and GR is the growth rate.

$$J_{dp}^{\pm} = \frac{dN_{dp}^{\pm}}{dt} + CoagS_{dp} \cdot N_{dp}^{\pm} + \frac{GR}{\Delta dp} \cdot N_{dp}^{\pm} + \alpha \cdot N_{dp}^{\pm} \cdot N_{<dp}^{\mp} - \chi N_{dp} \cdot N_{<dp}^{\pm} \qquad (4)$$

The fourth and fifth terms on the right hand side of equation (4) represent ion-ion recombination and charging of neutral particles by smaller ions, respectively, $\alpha$ is the ion-ion recombination coefficient and $\chi$ is the ion-aerosol attachment coefficient.

As per suggestions of both referees, the following discussion on atmospheric ions and ion induced nucleation as well as Figure R2-2, Figure R2-3 and Figure R2-4 were added to the manuscript as section 3.5.

3.5 Atmospheric ions and ion induced nucleation in Beijing

In order to estimate the contribution of ions to the total cluster mode particle number concentration and the importance of ion induced nucleation in Beijing, we studied ion number concentrations in the size range of 0.8-7 nm by dividing them into 3 sub-size bins: constant pool (0.8-1.5 nm), charged clusters (1.5-3 nm) and larger ions (3-7 nm). As shown in Figure R2-2, number concentrations of positive ions were higher than those negative ions in all the size bins on both NPF event days and haze days. We will only discuss positive ions here.

The median number concentration of positive ions in the constant pool on NPF event days was only 100 cm$^{-3}$ in Beijing, much less than that in the boreal forest (600 cm$^{-3}$; Mazon et al., 2016). Also, the median number concentration of positive charged clusters was 20 cm$^{-3}$ on the NPF event days, and the ratio to the total cluster mode particle number concentration was 0.001 to 0.004 during the NPF time window (Figure R2-3). This ratio is comparable to that observed in San Pietro Capofiume (0.004), in which the anthropogenic pollution level was also high, but clearly lower than that observed in another megacity in China, Nanjing (0.02; Kontkanen et al., 2017). Considerably higher ratios were observed in clean environments, for example during winter in the boreal forest at Hyytiälä, Finland (0.7; Kontkanen et al., 2017). The median number concentration of larger ions (3-7 nm) on the NPF event days was 30 cm$^{-3}$, a little bit higher than the charged cluster mode particle number concentration, indicating that not all of the larger ions originate from the growth of charged clusters, but rather from charging of neutral particles by smaller ions. On the haze days, charged ion number concentrations were much lower than those on the NPF days, which could be attributed to the higher condensation sink.

The diurnal pattern of the ratio of number concentration between charged and total cluster mode particles was the highest during the night with a maximum of 0.008, and had a trough during daytime with a minimum of 0.001 on the NPF event days. Such diurnal pattern is similar to earlier observations in Nanjing, San Pietro Capofiume and Hyytiälä (Kontkanen et al., 2017). This ratio reached its minimum around noon, because the total cluster mode particle number

concentration reached its maximum around that time due to NPF. The ratio had a small peak at around 9:00, similar to earlier observations in Centreville and Po Valley (Kontkanen et al., 2016;Kontkanen et al., 2017). The possible reason is that charged clusters were activated earlier in the morning than neutral clusters. The ratio increased from the midnight until about 4:00, similar to the number concentration of charged clusters.

As shown in Figure R2-4, the diurnal median of the ratio of formation rate of positive ions of 1.5 nm ($J_{1.5}^+$) to the total clusters of 1.5 nm ($J_{1.5}$) varied from 0.0009 to 0.006. This result is comparable to observations in Shanghai, where the positive ion induced nucleation contributed only 0.05% to the total formation rate of particles of 1.7 -nm ($J_{1.7}$) (Yao et al., 2018).

We updated our discussion in the manuscript from line 252 to line 259 with a more comprehensive study on traffic- related cluster and nucleation mode particles as below:

In Figure R2-4, we show the median diurnal pattern of particle number size distribution on the NPF event days and haze days separately. On the NPF event days, we observed cluster formation from diameters smaller than 3 nm. The growth of newly-formed particles lasted for several hours, resulting in a consecutive increase of the particle number concentrations in all the four modes. During traffic rush hours in the morning and evening, we observed an increase of particle number concentrations in the size range of cluster mode to around 100 nm.

On the haze days, we still observed an increase of particle number concentration in the size range of cluster mode to Aitken mode during rush hours. Traditionally, NPF events occur during the time window between sunrise and sunset by photochemical reactions (Kerminen et al., 2018). The binary or ternary nucleation between sulfuric acid and water, ammonia or amines are usually thought of as sources of atmospheric cluster mode particles, especially in heavily polluted environments (Kulmala et al., 2013;Kulmala et al., 2014; Yao et al., 2018; Chu et al., 2019). The burst of cluster mode particle number concentration outside the traditional NPF time window, especially during the rush hours in the afternoon, suggests a very different source of cluster mode particles from traditional nucleation, e.g. nucleation from gases emitted by traffic (Rönkkö et al., 2017).

As shown in Figure 6, on the NPF event days, the cluster mode particle number concentration started to increase at the time of sunrise and peaked around noon with a wide single peak, showing the typical behavior related to NPF events (Kulmala et al., 2012). Comparatively, on the haze days, the cluster mode particle number concentration showed a double peak pattern similar to the diurnal cycle of $NO_x$ (Figure 5). This observation in consistent with our discussion above that traffic emission possibly contributed to cluster mode particles. By comparing cluster mode particle number concentrations between the haze days and NPF event days, we estimated that traffic-related cluster mode particles could contribute up to 40-50 % of the total cluster mode particle number concentration on the NPF event days.

As per suggestion of the referee, the following discussion on growth rates of cluster and nucleation mode particles on NPF event days as well as Figure R2-5 were added to our manuscript as section 3.6:

3.6 Particle growth rates

The growth rates of particles generated from NPF events were examined in three size ranges: <3 nm, 3-7 nm and 7- 25 nm (Figure R2-5). The median growth rates of particles in these size ranges were 1.0 nm/h, 2.7 nm/h and 5.5 nm/h, respectively. The growth rate of cluster mode particles was comparable with that observed in Shanghai (1.5 nm/h; Yao et al., 2018). The notable increase of the particle growth rate with an increasing particle size is a very typical feature in the sub-20 nm size range (Kerminen et al., 2018), and it may also extend to larger particle sizes (Paasonen et al., 2018).

Our observations are in line with the reported range of nucleation mode particle growth rates of 0.1-11.2 nm/h in urban areas of Beijing (Wang et al., 2017;Jayaratne et al., 2017). Such growth rates can explain the observed increases of Aitken mode particle number concentrations in the afternoon.

Specific comments

Table 1: It would be useful for the reader if the authors mention in the text the number of days classified as NPF and haze.

Thank you for your suggestions.

We added in line 180 'We observed 28 NPF event days and 24 haze days' as per suggestion of the referee.

I don't see the definition of the modes. Also, which instrumentation did you use to calculate the different modes number concentrations? There are overlapping size ranges for nucleation and Aitken modes in the instrumentation you described.

The definition of the modes used in our study are introduced on lines 184-186: cluster mode (smaller than 3 nm), nucleation mode (3- 25 nm), Aitken mode (25- 100 nm) and accumulation mode (100- 1000 nm).

As per suggestion of the referee, the following discussion was added to our manuscript in line 186:

We calculated cluster mode particle number concentrations using Particle Size Magnifier (PSM) data, nucleation mode particle number concentration using Neutral Cluster and Air Ion Spectrometer (NAIS) particle mode data, and Aitken and accumulation mode particle number concentrations using Particle Size Distribution (PSD) system data.

The Particle Size Distribution system (PSD) and Neutral Cluster and Air Ion Spectrometer (NAIS) had an overlapping particle size distribution over the mobility diameter range of 3-42 nm. As shown in Figure R2-6, total particle number concentrations from the NAIS and PSD system correlated well with each other on both NPF event days ($R^2$ was 0.92) and haze days ($R^2$ was 0.90) in the overlapping size range. The slopes between the total particle number concentration from the PSD system and that from the NAIS were 0.90 and 0.85 on the NPF event days and haze days, respectively. The particle number size distribution in the overlapping size range of the NAIS and PSD system matched well on both NPF event days and haze days as shown in Figure R2-7.

Section 2.2: Please include the time resolution of the data you use. I could not find this

information for the PSD system and trace gases. How did you merge the data when the different instruments have different time resolutions? It is important that you describe this procedure carefully.

The following changes were made to our manuscript as per suggestion of the referee.

Line 145 as 'In the operation of the PSM, the saturator flow rate scanned from 0.1 to 1.3 lpm and scanned back from 1.3 to 0.1 lpm within 240 s. We averaged the data over 3 scans to make it smoother, and therefore the time resolution of PSM data was 12 minutes.'

Line 151 'the time resolution of PSD system data was 5 minutes'.

We added in line 165 that 'the time resolution of CO, $NO_x$, and $O_3$ data were 5 minutes, whereas the time resolution of $SO_2$ data was 1 hour before 22, January, 2018, and 5 minutes after that.' as per suggestion to the referee.

When data sets having different time resolutions were used, we chose the smallest time resolution as the common time resolution. Data with higher time resolutions were merged to the common time resolution by taking median numbers between two time points of the new time series.

Line 181: "In general, there were no overlap between NPF and haze periods". Did you look for NPF events during haze days? If haze and NPF are not 100% mutually excluding it would be interesting to describe these episodes. If they are, then change your sentence to make it clear that there was never an overlap. Also, did you determine haze days or did the China Meteorological Administration do this? If the authors did the classification, they should include the instrumentation used.

We thank the referee for the suggestions. We observed NPF events and haze events occurred on the same day, but never at the same time. We classified haze days not only according to the definition given by the China Meteorological Administration on the visibility and relative humidity but also the time haze lasts. Days were classified as haze days when haze events lasted for at least 12 consecutive hours. According to this classification, we did not observe any overlap between NPF event days and haze days.

As per suggestion of the referee, the following changes has been made in our manuscript:

We will update line 181 with 'NPF events and haze days as these two phenomena never occurred simultaneously.'

We added in section 2.2 'We measured relative humidity (RH, %) and, visibility (km), wind speed (m/s) and wind direction (˚) from a weather station on the roof of our station.'

Lines 222-228: The authors should also talk about $O_3$ here. I only see information for $SO_2$, CO and $NO_x$.

We thank the referee for the suggestions, we added the following description to our manuscript as per suggestion of the referee:

The median concentration of $O_3$ was 10 ppb on the haze days during our observations, a little bit higher than the severe haze episode in 2013 (<7 ppb; Wang et al., 2014b).

Lines 232-233: NPF does not favor clean environments. In any case, clean environments favor NPF.

We modified the corresponding text correspondingly.

Their lower levels on NPF event days indicates that relatively clean conditions favor NPF events.

Lines 247-251: "$NO_x$ and CO are important precursors of $O_3$ in Chinese urban areas. Based on our data, $O_3$, on the other hand, started to increase […] after the levels of $NO_x$ and CO started to decrease". Your wording is confusing. It feels like you are suggesting that $NO_x$ and CO are not precursors of $O_3$. Please reword.

We thank the referee for the suggestion, we rephrased the text as follows.

Earlier observations in urban areas having high $NO_x$ concentrations found that $O_3$ was consumed by its reaction with NO, while $NO_2$ works as precursor for $O_3$ via photochemical reactions (Wang et al., 2017). In our observations, the diurnal pattern of $O_3$ was opposite to that of $NO_x$, which is consistent with $O_3$ loss by large amounts of freshly emitted NO during rush hours and $O_3$ production by photochemical reactions involving $NO_2$ after the rush hours in the morning.

Lines 280-284: I don't see Aitken mode concentrations being similar to $NO_x$ evolutions before 9:00. See my comment on Figure 6 below.

We thank the referee for the suggestion. We modified Figure 6 as shown below. We can see Aitken mode number concentration increased during traffic rush time in Figure 6 after we changed the y-scale.

Lines 295-297: Did you measure meteorological parameters or is this a general statement? If it is a general statement change "the wind was" for "the wind is…".

We thank the referee for the comments. Yes, we measured meteorological parameters.

Line 301: It would be interesting to see the graphs for CS instead of giving only a daily value.

We thank the referee for the comments.

As per suggestion of the referee, we added Figure R2-8 to our manuscript. Figure R2-8 describes the condensation sink on both NPF event days and haze days.

Section 3.3: In line 318 the authors state "In this section, we use CO, $SO_2$, $NO_x$ and $O_3$ as tracers", but there are no comments whatsoever regarding CO or $O_3$ in this section.

We thank the referee for the comments. Combined with the comments on Table 2, we updated discussion at section 3.3 and added the discussion about the limitation on the method in line 319 as below:

[revised manuscript text omitted]

Line 337: Looking at the figure, it doesn't seem to me that $SO_2$ and cluster and nucleation mode concentrations are correlated, especially for NPF days. What are the correlation coefficients for NPF days and haze days separately? (see also comment for Table 2).

We thank the referee for the comments. Although $SO_2$ is precursor of sulfuric acid and we would have expected a positive correlation with cluster mode particles, there are other species involved such as $NH_3$, dimethyl amine (DMA), that could limit the process.

As per suggestion of the referee, we will update our discussion in our manuscript as shown in the response to the last comments above.

Lines 400-402: Please elaborate and comment on the correlation with the other modes. PM$_{2.5}$ is also highly correlated with cluster mode but this is not discussed. Consider showing the correlations in an additional figure.

We thank the referee for the comments. We added correlation coefficient between PM$_{2.5}$ concentration and every mode on NPF event days and haze days in Table R2a and Table R2b separately. The following discussions and Figure R2-9 were added to our manuscript as section 3.3.5 as per suggestions to the referee:

3.3.5 Connection to PM$_{2.5}$

As shown in Figure R2-9, PM$_{2.5}$ concentration correlated negatively with the cluster and nucleation mode particle number concentrations, and positively with the accumulation mode particle number concentration. High PM$_{2.5}$ concentrations tend to suppress NPF by increasing the sinks of vapors responsible for nucleation and growth of cluster and nucleation mode particles. The particles causing high PM$_{2.5}$ concentrations also serve as sinks of cluster and nucleation mode particles by coagulation.

As shown in Table R2a and Figure R2-9, the Aitken mode particle number concentration correlated positively with the PM$_{2.5}$ concentration on the NPF event days. A possible reason for this could be the tight connection between the Aitken and accumulation mode particles on the NPF event days (Table R1a), and the observation that accumulation mode particles are usually the main contributor to PM$_{2.5}$ in Beijing (Liu et al., 2013). On the haze days, the Aitken mode particle number concentration correlated negatively with the PM$_{2.5}$ concentration (Table R2b). A possible reason for this is that pre-existing large particles acted as a sink for Aitken mode particles by coagulation as well as a sink for vapors responsible for the growth of smaller particles into the Aitken mode. In addition, while PM$_{2.5}$ is dominated by regional and transported secondary aerosols, Aitken mode particles mainly originate from local emissions such as traffic and cooking in Beijing (Wu et al., 2007;Wang et al., 2013;Du et al., 2017;de Jesus et al., 2019).

Conclusion: This section is written as a summary. The conclusion should reflect the significance of the results presented in this paper compared with existing observations, and give a message beyond summarizing what has already been said in the previous sections.

We thank the referee for the comments. we revised our conclusion as per suggestion of the referee as below:

4. Summary and conclusions

We measured particle number concentrations over a wide range of particle diameters (1.5-1000 nm) on both NPF event days and haze days in winter Beijing. To our knowledge, this was the first time when cluster mode particle number concentrations have been reported on haze days in Beijing.

The observed responses of particle number concentrations in different modes (cluster, nucleation, Aitken and accumulation mode) to changes in trace gas and PM$_{2.5}$ concentrations were quite heterogeneous, suggesting different sources and dynamics experienced by each mode. NPF was the dominant source of cluster and nucleation mode particles. Ion-induced

nucleation did not play an important role during the NPF events. The growth rates of cluster and nucleation mode particles increased with an increasing particle size. Traffic emissions contributed to every mode and were the dominant source of cluster and nucleation mode particles on the haze days. The main sources of Aitken mode particles were local emissions, while transported and regional pollution as well as growth from the nucleation mode also contributed to the Aitken mode. The main source of accumulation mode particles was regional and transported pollution. $PM_{2.5}$ affected the number concentration of sub-100 nm particles by competing for vapors responsible for particle growth and by acting as sinks for particles by coagulation. The main contributors to the $PM_{2.5}$ mass concentration were accumulation mode particles on the haze days.

As demonstrated here and in many other studies (Brines et al., 2015), ultrafine particles (< 100 nm in diameter) tend to dominate the total aerosol particle number concentration in megacities like Beijing. We should put no less attention on ultrafine particles than larger particles for their large number population. More attention should therefore put on ultrafine particles in urban environments. We found that both NPF and traffic emissions are important sources of ultrafine particles in Beijing. To improve our understanding on the potential effects of ultrafine particles on health and air quality, we need to do more research on their sources and physical and chemical properties. Laboratory and model analysis on dynamics of ultrafine particles would help us to understand the evolution of particle number size distributions. In addition, to identify and locate other possible sources, long-term observations on ultrafine particles down to the cluster mode as well as source apportionment analyses, such as cluster analysis and receptor model studies, are still needed. Ultrafine particles should also be taken into consideration when making policies to control air pollution. New regulations should be designed to control primary emission sources, such as traffic, or precursor emissions for secondary ultrafine particles involving NPF and subsequent particle growth.

Line 415-416: I do not see in the text where this is discussed (secondary sources contribution to the Aitken mode during haze days).

We thank the referee for the comments. We updated our conclusion and we deleted this sentence about 'secondary sources contribution to the Aitken mode during haze days' in our conclusions.

Table 2: I think this table would be more useful if the authors separate the data for haze days and NPF days. Also, what do you mean by "all the data are in log scale"? Please reword. You are showing correlation coefficients here, which are not represented in any scale.

We deleted the words 'all the data are in log scale'.

As per suggestion of the referee, we separated the data for NPF and haze days by changing Table 2 into Table R2a and Table R2b as shown below.

Figures 1, 2, 3, 4: Are these daily averages? Please specify the data you used to make the plot.

Data in Figure 1,3,4 are raw data we observed. Data in Figure 2 are the median of all the raw data of each mode on NPF event days, haze days and others days separately

Figures 5 and 6: Please specify the time resolution of the data you are showing, and reword "and they are the median data from midnight to midnight".

We thank the referee for the comments.

As per suggestion of the referee, we changed captions of Figure 5 and 6 as below:

Figure 5. Diurnal variation of trace gas (CO, $SO_2$, $NO_x$ and $O_3$ separately) mixing ratios on the NPF event days (green lines) and haze days (grey lines) separately. The time resolution was 30 minutes for every data point. Every data point here represents the median of all data at the same time of the days.

Figure 6. Diurnal variation of particle number concentration of every mode (cluster, nucleation, Aitken and accumulation mode separately) on the NPF event days (green lines) and haze days (grey lines). The time resolution was 30 min for every data point. Every data point here represents the median of all data at the same time of the days.

Figure 5: The scale in the upper left graph is different to the others. If you decide to use the same scale, change it to match the others. If it is not important for you that the graphs have the same scale, change the other scales (especially $SO_2$ and $O_3$) so that the variations can be seen more clearly.

We thank the referee for the comments.

As per suggestion of the referee, to make the variations to be seen more clearly, we changed the scales as shown in Figure 5 below.

Figure 6: The scales used here do not allow to see changes in the Aitken and accumulation modes. I would suggest changing the scales on the lower graphs. It is hard to see the changes you mention in the discussion.

As per suggestion of the referee, to make the variations to be visible, we changed the scales as shown in Figure 6 below.

Technical comments

Line 93-94: "…. complicating the story even further". I would suggest using a different language.

We thank the referee for the comments. We will update the description in our manuscript as below:

While cluster mode particles can grow into the Aitken mode , also other sources like traffic contribute to this mode, making the source identification of the Aitken mode complicated.

Line 153: Please change the verb tense: measures -> measured.

As per suggestion of the referee, we corrected the verb tense.

Lines 169 and 175: Correct the references format.

As per suggestion of the referee, we corrected the reference format.

Line 213-214: Check the Aitken and accumulation median concentrations. Are they exactly the same?

We corrected the median number concentration of Aitken mode on haze days as 16000 cm$^{-3}$ in our manuscript.

Line 300: Change "maybe" for "may be".

As per suggestion of the referee, we changed 'maybe' for 'may be' in our manuscript in line 300.

Line 305: Delete "in" between "increase" and "during".

As per suggestion of the referee, we deleted "in" between "increase" and "during" in line 305.

Line 311: Add in: "… $SO_2$ participated in the formation…"

As per suggestion of the referee, we added in: "… $SO_2$ participated in the formation…"

Line 386: "resulting in an increase…"

As per suggestion of the referee, we corrected our words as "resulting in an increase…"

Figure 2: Switch "left" and "right" in the figure caption.

As per suggestion of the referee, we switched "left" and "right" in the figure caption.

Figure 3: Please change the label: OtherS -> Others.

As per suggestion of the referee, we changed the label from "OtherS" to "Others" in Figure 3.

Tables and Figures

[revised manuscript text omitted]

Wu, Z. J., Hu, M., Liu, S., Wehner, B., Bauer, S., Ssling, A. M., Wiedensohler, A., Petaja, T., Dal Maso,

M., and Kulmala, M.: New particle formation in Beijing, China: Statistical analysis of a 1-year data set, J Geophys Res-Atmos, 112, 2007.

Yao, L., Garmash, O., Bianchi, F., Zheng, J., Yan, C., Kontkanen, J., Junninen, H., Mazon, S. B., Ehn, M., Paasonen, P., Sipilä, M., Wang, M. Y., Wang, X. K., Xiao, S., Chen, H. F., Lu, Y. Q., Zhang, B. W., Wang, D. F., Fu, Q. Y., Geng, F. H., Li, L., Wang, H. L., Qiao, L. P., Yang, X., Chen, J. M., Kerminen, V. M., Petäjä, T., Worsnop, D. R., Kulmala, M., and Wang, L.: Atmospheric new particle formation from sulfuric acid and amines in a Chinese megacity, Science, 361, 278-281, https://doi.org/10.1126/science.aao4839, 2018.